# Membrane manipulation by free fatty acids improves microbial plant polyphenol synthesis

Apilaasha Tharmasothirajan [1,2], Josef Melcr[3], John Linney[4], Thomas Gensch [5], Karin Krumbach[1], Karla Marlen Ernst[1], Christopher Brasnett [3], Paola Poggi[6], Andrew R. Pitt[4,7], Alan D. Goddard[4], Alexandros Chatgilialoglu[6], Siewert J. Marrink [3] & Jan Marienhagen [1,2] ✉

Microbial synthesis of nutraceutically and pharmaceutically interesting plant polyphenols represents a more environmentally friendly alternative to chemical synthesis or plant extraction. However, most polyphenols are cytotoxic for microorganisms as they are believed to negatively affect cell integrity and transport processes. To increase the production performance of engineered cell factories, strategies have to be developed to mitigate these detrimental effects. Here, we examine the accumulation of the stilbenoid resveratrol in the cell membrane and cell wall during its production using *Corynebacterium glutamicum* and uncover the membrane rigidifying effect of this stilbenoid experimentally and with molecular dynamics simulations. A screen of free fatty acid supplements identifies palmitelaidic acid and linoleic acid as suitable additives to attenuate resveratrol's cytotoxic effects resulting in a three-fold higher product titer. This cost-effective approach to counteract membrane-damaging effects of product accumulation is transferable to the microbial production of other polyphenols and may represent an engineering target for other membrane-active bioproducts.

Secure access to plant-derived polyphenols such as flavonoids and stilbenoids is of great importance for the food, feed, cosmetic and pharmaceutical industries[1,2]. This interest can be largely attributed to the natural functions of stilbenoids as antioxidants, antibiotics or radical scavengers in plants[3,4]. The best known stilbenoid is resveratrol (RES), which shows a broad range of health-promoting anti-diabetic, anti-cancerogenic and anti-inflammatory properties in pre-clinical studies[5,6]. In principle, polyphenols such as RES can be extracted from natural plant producers or chemically synthesized from compounds stemming from the BTEX-fraction of petroleum-derived feedstocks. However, due to their generally low abundance

in plants or the requirement for chemical precursors and organic solvents for chemical synthesis, both options are usually not considered to be sustainable[7]. In this context, microbial synthesis represents a promising alternative for the production of valuable plant polyphenols starting from abundant and inexpensive sugar-based feedstocks[8].

In the past decades, microorganisms such as *Saccharomyces cerevisiae* and *Escherichia coli* have been transformed into microbial cell factories for plant polyphenol production[9,10]. More recently, the Gram-positive soil bacterium *Corynebacterium glutamicum*, used for the industrial amino acid production at a multi-million ton scale, was

[1]Institute of Bio- and Geosciences, IBG-1: Biotechnology, Forschungszentrum Jülich, 52425 Jülich, Germany. [2]Institute of Biotechnology, RWTH Aachen University, Worringer Weg 3, 52074 Aachen, Germany. [3]Groningen Biomolecular Sciences and Biotechnology Institute, University of Groningen, 9747 AG Groningen, The Netherlands. [4]College of Health and Life Sciences, Aston University, Birmingham B4 7ET, UK. [5]Institute for Information Processing, IBI-1: Molecular and Cellular Physiology, Forschungszentrum Jülich, 52425 Jülich, Germany. [6]Remembrane Srl, via San Francesco 40, 40026 Imola, Italy. [7]Manchester Institute of Biotechnology and Department of Chemistry, University of Manchester, Manchester, UK. ✉e-mail: j.marienhagen@fz-juelich.de

engineered for the synthesis of plant polyphenols[11]. The identification and elimination of a catabolic pathway for phenylpropanoids as polyphenol precursors and increased intracellular availability of malonyl-CoA, which is also required for polyphenol synthesis, were important prerequisites for establishing *C. glutamicum*-based polyphenol-producing cell factories (Fig. 1)[12–15].

Despite the robustness of *C. glutamicum* against elevated concentrations of aromatic compounds in comparison to other micro-

**Fig. 1 | Schematic representation of the pathway for the synthesis of RES and NAR.** AroH 3-deoxy-D-arabino-heptulosonate-7-phosphate synthase, TAL tyrosine ammonia lyase, 4CL 4-coumarate:CoA ligase, STS stilbene synthase, CHS chalcone synthase, CHI chalcone isomerase.

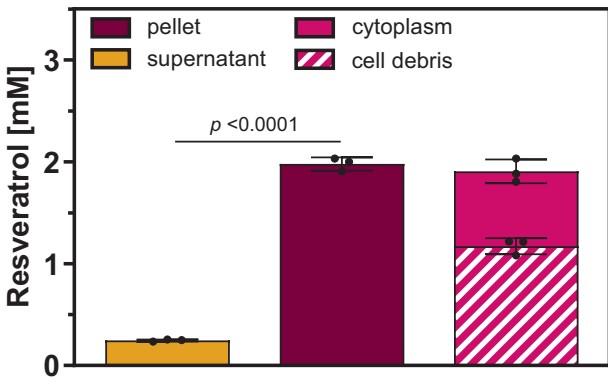

**Fig. 2 | Localization of RES in *C. glutamicum*-RES1.** Quantification of RES in the culture supernatant (yellow), cell pellet (magenta), cytoplasm (light magenta), and membrane fraction (striped magenta) after RES production in *C. glutamicum*-RES1. Data represent average values and standard deviation of three biological replicates (*n* = 3). Statistical significance was calculated by a two-tailed unpaired Student's *t* test. Source data are provided as a Source data file.

**Table 1 | Investigated FA supplements**

| Fatty acid | C:D |
|---|---|
| Lauric acid | 12:0 |
| Myristic acid | 14:0 |
| Arachidic acid[a] | 20:0 |
| Lignoceric acid | 24:0 |
| Lauroleic acid[a] | 12:1n1 |
| Myristoleic acid | 14:1n5 |
| Palmitoleic acid | 16:1n7 |
| Oleic acid | 18:1n9 |
| Nervonic acid[a] | 24:1n9 |
| Linoleic acid | 18:2n6 |
| Palmitelaidic acid[a] | t16:1n7 |
| Linoelaidic acid[a] | tt18:2n6 |

"C" defines the number of carbon atoms and "D" the number of double bonds in the carbon chain.
[a]Not native to *C. glutamicum*.

organisms, the antibacterial properties of RES, as with many other polyphenols, renders the microbial synthesis of these valuable molecules challenging[16,17]. This inherent product toxicity towards the producing cells remains a great challenge on the way to a microbial production of these compounds at larger scale[18]. Application of organic solvents as a second phase during microbial polyphenol production represents an interesting possibility to mitigate product toxicity. Recently, it was shown that implementation of such a biphasic extractive cultivation process reduces the cytotoxic effect of RES towards the producing *C. glutamicum* cells, which increased productivity and product yield[19]. However, product recovery from solvents used for extraction is typically energy intensive and often associated with high consumption of acids and bases[20].

Interestingly, the cytotoxic effects of RES on microbial cells have been only investigated to a limited extent, and only a detrimental effect of RES on DNA cleavage and impairment of cell division has been described in detail[21]. However, due to its physical properties it has been proposed that RES preferentially accumulates on and in the cytoplasmic membrane, which leads to the impairment of membrane integrity and vital cell functions[22–24].

Recent studies demonstrated that bacteria such as *Enterococcus faecalis* or pathogenic *Vibrio* species are able to incorporate exogenous fatty acids (FA) into membrane phospholipids, thereby altering their membrane composition to increase antibiotic resistance[25,26]. However, potentially beneficial effects of exogenously supplied FAs on microbial membrane characteristics in the context of biotechnological production processes have not been extensively studied.

In this work, we investigate the impact of RES on the membrane of *C. glutamicum* and show that supplementation of small amounts of selected FAs can be used to effectively attenuate toxic effects of polyphenols, and boost productivity and overall product yield. This demonstrates that exogenous addition of free FAs represents a viable strategy to increase tolerance to a range of membrane-active bioproducts.

## Results
### Localization of resveratrol in *C. glutamicum*
*C. glutamicum*-RES1 engineered for RES production synthesizes up to 0.75 ± 0.1 mM RES in shake flasks within 72 h, which was verified by a total extraction of the cultures with ethyl acetate and subsequent HPLC-MS analysis[19]. However, in these cultivations typically 15% of the supplemented phenylpropanoid precursor *p*-coumaric acid (5 mM, *p*-CA) was converted to RES, indicating a compromised production

process. Separate analysis of the culture supernatant showed that less than 35% (0.26 ± 0.01 mM) of the synthesized RES is present in the culture supernatant, whereas the majority of the product appeared to accumulate in the cells[19]. This finding was remarkable considering that the biomass typically constitutes less than 3% of the culture volume. For a more detailed analysis, product accumulation in supernatants and cells during cultivations of *C. glutamicum*-RES1 was followed over time. In relation to the individual volumes of culture supernatant and cell pellet, a maximum concentration of 0.25 ± 0.01 mM RES could be quantified in the culture medium after 72 h, whereas an 8-fold higher RES-concentration (2 ± 0.05 mM) was determined in the cell pellet (Fig. 2). Motivated by these findings, RES was separately quantified in the cytoplasm (0.72 ± 0.01 mM) as well as membrane and cell wall (1.2 ± 0.03 mM) as insoluble cellular components (cell debris) of *C. glutamicum*-RES1. This corresponds to approximately 30% and 58% of the total product concentration, respectively. Hence, RES appears to be mainly located in the membrane fraction of *C. glutamicum*-RES1 and it can be assumed that such high product concentrations in and on the cells have a negative effect on cellular functions, eventually limiting overall productivity. Previous studies also demonstrated a dose-dependent inhibitory effect of different RES concentrations on the growth of *C. glutamicum*. In the context of these experiments, an $IC_{50}$ of 2.6 mM could be determined for RES[19].

### Screening of fatty acid supplements to reduce resveratrol cytotoxicity
Extracellular FA supplementation to modify the cell membrane composition and properties is largely unexplored in prokaryotic Gram-positive organisms. However, the previously discussed finding that supplementation of FAs can positively influence the membrane composition of some bacteria for enhanced antibiotic resistance, motivated the individual external supplementation of twelve different FAs to producing *C. glutamicum*-RES1 cultures with the aim to improve RES efflux across the cell envelope[25,26]. In initial cultivation experiments, biomass formation and RES production of *C. glutamicum*-RES1 in the presence of a representative sample of FAs (4 saturated FAs, 6 *cis*-unsaturated FAs, 2 *trans*-unsaturated FAs, Table 1) was investigated. Seven of these FAs (12:0, 14:0, 24:0, 14:1n5, t16:1n7, 18:1n9, 18:2n6) are native to *C. glutamicum* but are only present in membrane phospholipids in very low amounts[27].

The total RES concentration was quantified in the culture (cells and supernatant) and in the culture supernatant only. Since FAs are solubilized in ethanol (EtOH), a second control containing the same final EtOH concentration as the FA-supplemented cultures was included. Noteworthy, increased biomass formation was observed in the

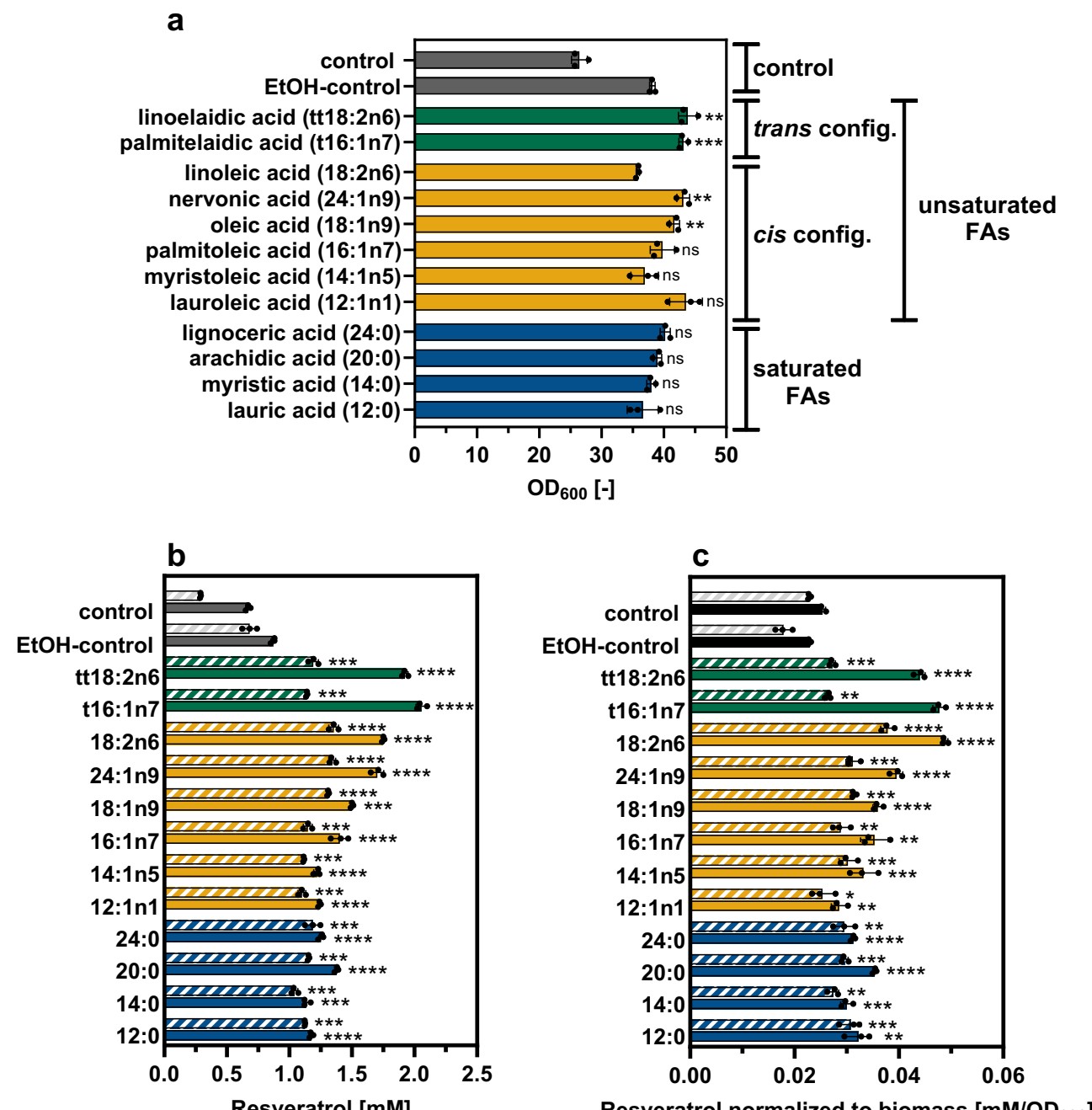

**Fig. 3 | FA supplementation for improved RES production. a** Final biomass and **b** product titer as well as **c** normalized RES concentrations to biomass of *C. glutamicum*-RES1 in presence of different FA supplements. In all figures showing product concentration, filled bars represent total RES, whereas striped bars represent extracellular RES. Data represent average values and standard deviations of three biological replicates ($n = 3$). Statistical significance between the experimental groups and the EtOH-control for total RES and extracellular RES was assessed using a two-tailed unpaired Student's *t* test (****$p < 0.0001$, ***$p < 0.001$, **$p < 0.01$, *$p < 0.05$, ns not significant). Source data are provided as a Source data file, including exact *p*-values.

EtOH-containing controls compared to the control flasks without EtOH, since *C. glutamicum* is able to metabolize EtOH (Fig. 3a)[28]. Supplementation of saturated FAs (12:0, 14:0, 20:0, 24:0) and unsaturated *cis*-FAs (12:1n1, 14:1n5, 16:1n7, 18:1n9, 24:1n9, 18:2n6), did not result in a significant increase in final biomass compared to the EtOH control. Nevertheless, supplementation of unsaturated *trans*-FAs (t16:1n7 and tt18:2n6) apparently increased the biomass formation by 14%. In case of these two *trans*-FAs, metabolization by the microorganism could be an explanation, but *C. glutamicum* ATCC 13032 does not have a β-oxidation pathway for FA catabolism. Hence, growth experiments in the presence of 40 μM of each of the 12 FA

supplements as sole carbon and energy source were conducted to investigate a possible metabolization of any of these compounds (Supplementary Fig. 1). In all cases the growth behavior was similar to the EtOH control confirming that *C. glutamicum*-RES1 cannot metabolize any of the supplemented compounds. Interestingly, the addition of all 12 FA supplements led to a significant increase in overall final RES titers and an increased extracellular RES accumulation in the supernatant compared to the EtOH control (Fig. 3b). Of all tested supplements, the highest RES titer was achieved upon supplementation of the non-native *trans*-unsaturated t16:1n7 (palmitelaidic acid: 2.1 ± 0.03 mM), tt18:2n6 (linoleaidic acid: 1.92 ± 0.02 mM) as well as the

native *cis*-unsaturated 18:2n6 (linoleic acid: $1.75 \pm 0.01$ mM). Indeed, RES concentrations in the culture supernatant were close to the total product concentration, a first indication for enhanced RES export.

RES production in *C. glutamicum*-RES1 is a growth-coupled production process. For a better comparison of the effects of the individual FA supplements, final product titers were normalized to optical density ($OD_{600}$) to obtain the individual product yields (Fig. 3c). This calculation showed that 41% more RES was produced in the presence of palmitelaidic acid (t16:1n7, tPAL) and linoleic acid (18:2n6, LA), respectively (tPAL/LA: $0.048 \pm 0.001$ mM/$OD_{600}^{-1}$), despite lower biomass. Based on these results, tPAL and LA supplementation was further investigated regarding the respective impact on growth, RES production, RES export across the membrane and cell wall.

## Supplementation of palmitelaidic acid or linoleic acid improves microbial resveratrol synthesis

Different concentrations of tPAL and LA were supplemented to *C. glutamicum*-RES1 cultivations (10; 20; 30; 40; 60; 90; 120; 180; 240 µM) to investigate the optimal FA concentration for improved RES production. Exposure of the production strain to different concentrations of *cis*-unsaturated LA demonstrated a dose-dependent growth inhibition by this FA (Supplementary Fig. 2). The addition of 10 µM LA demonstrated no impact on biomass formation and RES production compared to the EtOH-control. In contrast, the addition of 240 µM LA resulted in a 50% lower final $OD_{600}$ after 72 h of cultivation. A two-fold higher RES concentration was demonstrated in the approach with 40 µM LA (EtOH-control: $0.92 \pm 0.01$ mM; 40 µM LA: $1.75 \pm 0.01$ mM), whereas a decrease in total RES concentration was observed with increasing LA concentrations from 60 µM onwards, which can be attributed to the observed lower biomass concentrations. Nevertheless, in all cultivations with 40 µM LA more than 1.3 mM RES was produced in total despite lower biomass formation, resulting in an improvement of more than 50%. In comparison, addition of different tPAL concentrations did not have such a pronounced detrimental effect on *C. glutamicum*-RES1 growth, indicating a lower cytotoxicity of tPAL compared to LA (Supplementary Fig. 2). In case of tPAL, the highest RES titer was achieved upon supplementation of 40 µM tPAL ($1.94 \pm 0.03$ mM). Based on these results, 40 µM LA or 40 µM tPAL appear to be sufficient for increased RES production using *C. glutamicum*-RES1.

For a more detailed analysis, growth and RES production using *C glutamicum*-RES1 in the presence of 40 µM LA or tPAL was followed over time (Supplementary Fig. 3). Despite similar growth in the presence of LA and tPAL compared to the EtOH control, significant differences were observed in terms of RES production. In the absence of LA or tPAL, only low RES productivity could be detected in the control culture after 24 h (total: $0.02 \pm 0.001$ mM; supernatant: 0 mM). At the end of the cultivation, a total of $0.87 \pm 0.02$ mM RES was synthesized, of which $0.25 \pm 0.03$ mM RES was located in the culture supernatant. In contrast, supplementation of LA and tPAL enabled the accumulation of $0.1 \pm 0.01$ mM RES in the culture supernatant (and of $0.31 \pm 0.01$ mM RES in total) 24 h after inoculation. Similar product titers of $1.71 \pm 0.1$ mM RES (total) could be determined for both cultures with FA supplements after 48 h, a value that did not change until the end of cultivation. In order to explore the possibility of further increasing RES production in *C. glutamicum*-RES1, LA and tPAL were added simultaneously to the culture medium (Supplementary Fig. 3). An FA-mixture containing 20 µM LA and 20 µM tPAL (total FA concentration of 40 µM) and a mixture containing 40 µM LA and 40 µM tPAL (total FA concentration of 80 µM) were used. The addition of a 40 µM FA mixture resulted in the same final titer of 1.8 mM RES compared to the single supplementation with 40 µM LA or tPAL. However, the supplementation of the 80 µM FA mixture did not further improve the production performance.

## Resveratrol affects structure and properties of lipid membranes

External supplementation of tPAL and LA allowed for an increased RES production with *C. glutamicum*-RES1. However, since metabolization of both FAs can be excluded, the impact of RES, LA and tPAL on the membrane of *C. glutamicum* was investigated in more detail. Possible alterations in membrane fluidity during RES production were initially studied by Laurdan fluorescence assays. The generalized polarization (GP) value of Laurdan-labeled cells as a measure for membrane fluidity was calculated to evaluate the membrane fluidity of *C. glutamicum*-RES1 cells compared to cells in absence of RES (Fig. 4a)[29]. A GP value of $0.27 \pm 0.1$ was determined for *C. glutamicum*-RES1 at the beginning of the cultivation. During cultivation of *C. glutamicum*-RES1 without induction of heterologous gene expression for RES synthesis, the GP value increased only slightly to $0.29 \pm 0.1$ within 72 h, indicating no major changes in membrane fluidity in absence of RES. In contrast, for RES-producing *C. glutamicum*-RES1 cells, the GP value increased to $0.39 \pm 0.1$ over the course of cultivation, suggesting that RES exhibits a rigidifying effect on the membrane.

To investigate the immediate impact of extracellular RES on the membrane of *C. glutamicum*-RES1 Laurdan kinetic measurements were conducted (Fig. 4b and Supplementary Fig 4). For this, the emission of Laurdan-stained *C. glutamicum*-RES1 cells was followed for 10 min before 0.5 mM RES was added to the stained cells. In addition, the same amount of dimethyl sulfoxide (DMSO) (0.1% (v v$^{-1}$)) and EtOH (0.05% (v v$^{-1}$)) was added to the control to exclude interference in kinetic measurements. The fluorescence emission was followed for 60 min after the addition of RES or DMSO/EtOH. In the control culture, the GP value remained constant at $0.27 \pm 0.1$, despite the addition of DMSO and EtOH after 10 min of incubation. Compared to the control, the addition of 0.5 mM RES showed an increased GP value from $0.27 \pm 0.1$ to $0.58 \pm 0.1$ within 6 and 14 min, respectively. Further investigation demonstrated that the addition of 0.13 mM RES did not result in any discernible alteration in the membrane fluidity of *C. glutamicum*-RES1 (Supplementary Fig. 4). However, increasing extracellular RES concentrations ranging from 0.25–0.75 mM revealed a dose-dependent increase in GP values. These results indicate that the presence of RES leads to an immediate rigidifying effect on the membrane of *C. glutamicum*-RES1.

Fluorescence microscopy measurements were performed with Laurdan-stained *C. glutamicum* cells in presence and absence of RES to investigate its effects on the cell membrane. In previous studies, dipolar relaxation[30–32] (i.e., a spectral shift of Laurdan's fluorescence) as well as fluorescence properties[33–35] (i.e., fluorescence lifetime and fluorescence anisotropy) have been used to study membrane properties, such as membrane fluidity. In this context, it could be shown that it is advantageous to detect and analyze the blue and green portion of Laurdan's emission separately, when using Laurdan's fluorescence lifetime as readout parameter for membrane fluidity[30]. The blue channel mainly reflects the influence of the polarity of the dye's environment on the Laurdan ground state and changes caused by variation of, e.g., water or cholesterol are well reflected in the blue channel. Thus, the same excitation wavelength and spectral window (400–490 vs. 420–500 nm)[30] were used here to investigate the influence of RES on the fluorescence lifetime of Laurdan. After Lauran staining, the stained cell membranes of *C. glutamicum* were easily observable in fluorescence microscopic measurements (Fig. 5a and b). The fluorescence lifetime image (FLIM) is significantly shortened by the presence of RES (Fig. 5d), as indicated by a greener color in FLIMs compared to Laurdan-stained cells without RES (Fig. 5c). Due to the presence of RES, a decrease of approximately 16% in the fluorescence lifetime of Laurdan could be observed in fluorescence lifetime images (Fig. 5e). The shorter fluorescence lifetime of Laurdan in presence of RES was detected in three independent measurements with high statistical significance.

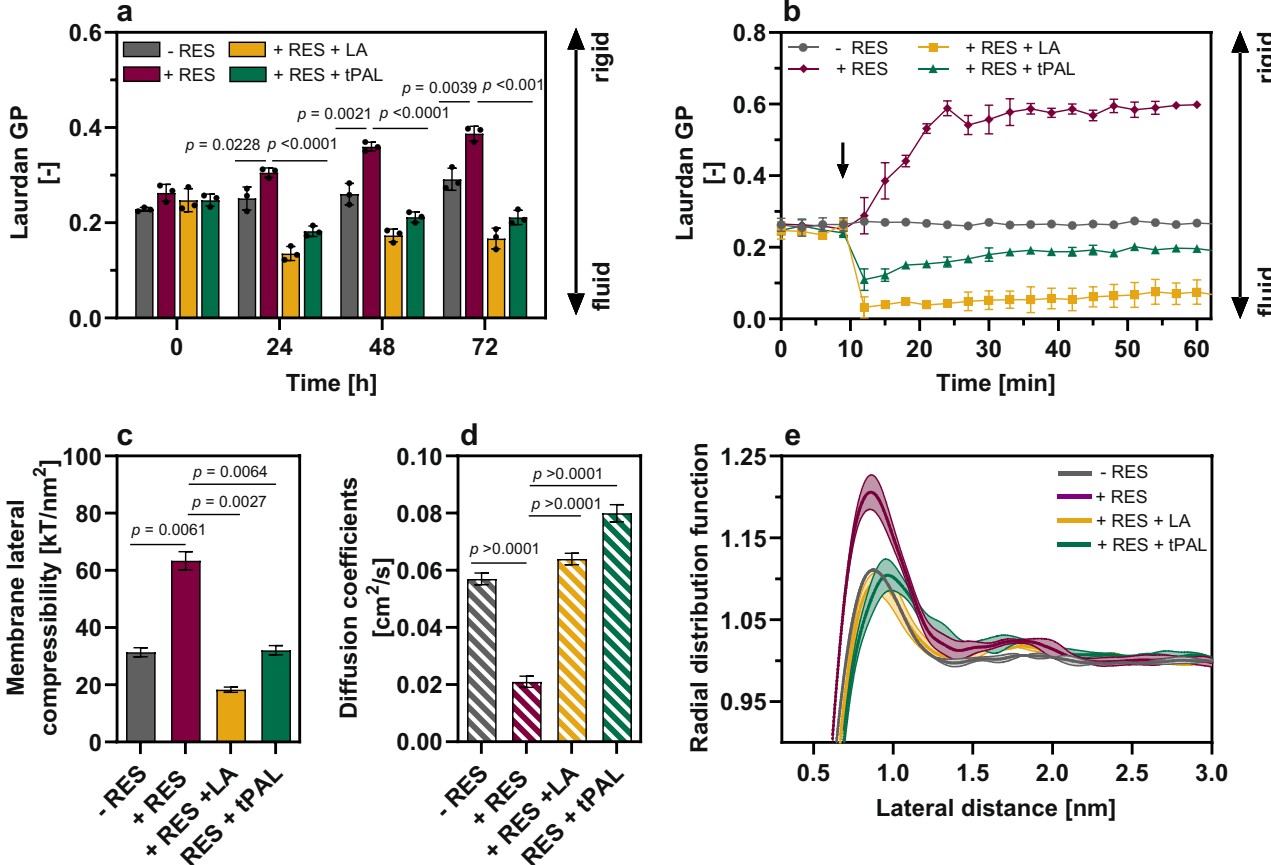

**Fig. 4 | Membrane fluidity of *C. glutamicum*-RES1 in presence or absence of RES and FAs. a** One-point measurements of Laurdan GP of *C. glutamicum*-RES1 cells in absence of RES (grey), in presence of RES (magenta) as well as in presence of RES and LA (yellow) and tPAL (green). Laurdan measurements were taken at an OD$_{600}$ of 0.5. Laurdan GP values at 0 h give the generalized polarization at the beginning of the cultivation in absence of LA or tPAL. **b** Kinetic Laurdan measurements with stained *C. glutamicum*-RES1 cells at OD$_{600}$ of 0.5. After 10 min, 0.5 µM LA, tPAL and 0.5 mM RES were added to the cells. The same volume of EtOH (0.05 (v v$^{-1}$) %) and DMSO (1 (v v$^{-1}$) %) was added as a control to exclude possible interferences by increasing the volume. Data of each graph represent average values and standard

deviations from three biological replicates (*n* = 3). **c** Membrane lateral compressibility and **d** diffusion coefficient from simulated membranes of different compositions. Block averaging method was used for the estimation of standard deviation. **e** 2D Radial distribution function of the POPG lipids at the membrane surface. Data from two membrane leaflets were taken as two independent measurements (*n* = 2). The solid line represents the average value, and the shaded areas indicates the standard deviation of two independent measurements. Statistical significance was calculated by a two-tailed unpaired Student's *t* test. Source data are provided as a Source Data file, including exact *p*-values.

To provide a molecular level interpretation of the increased GP-value from fluorescence measurements, MD simulations of a model *C. glutamicum* lipid membrane were performed using the coarse-grain Martini forcefield[36]. Based on the lipid analysis of *C. glutamicum* membranes, the most abundant negatively charged 1-palmitoyl, 2-oleoyl-sn-glycero-3-phosphatidylglycerol (POPG (50 mol%)) and 1,2-dipalmitoyl phosphatidylinositol (DPPI (25 mol%)) lipids were selected to model the *C. glutamicum* membrane, as well as 1-palmitoyl, 2-oleoyl-monogalactosyldiacylglycerol ((PO)MGDG (12.5 mol%)) and 1-palmitoyl, 2-oleoyl diacylglycerol (PODG (12.5 mol%)), which represents the remaining mix of overall neutral polar lipids[37,38]. To probe the effect of RES, 50 mol% RES was included. The available experimental and theoretical evidence demonstrated that RES has a high affinity to the surface of lipid bilayers in general[22,23,39–41], which agrees with our results for the *C. glutamicum* model membrane (Supplementary Fig. 5). A direct comparison of RES concentrations with the experimental setup is difficult, as the aqueous concentration is not very informative on the amount of RES bound to the membrane, given the strong affinity of RES for the latter. During the simulations, RES molecules were observed to detach from the membrane with very low probability, subsequently spending some time in the aqueous environment before rebinding.

Concordantly with the results of membrane fluorescence measurements, MD simulations show that RES significantly increases the rigidity of lipid membranes while slowing down their dynamics. This is reflected in the membrane lateral compressibility coefficient as measure for the elasticity of lipid membranes, which increases two-fold upon adding RES to a RES:lipid ratio of 1:2 (Fig. 4c and d and Supplementary Fig. 5). At the same time, the diffusion coefficient of POPG as the dominant lipid in *C. glutamicum* membranes reveals an approximately two-fold slowdown. The membrane rigidifying effect of RES is presumably linked to an alteration in the lateral organization of the membrane. The addition of RES brings the most abundant POPG lipids closer in proximity and increases their segregation from the other lipids, causing the formation of transient nanoscale lipid domains (Supplementary Fig. 5 and Supplementary Movies 1 and 2). This effect is shown by an increased probability of close-contact in 2D radial distribution function of POPG lipids after addition of RES (+RES) compared to the reference (−RES) (Fig. 4e and Supplementary Fig. 5). Interestingly, previous MD studies probing the effect of RES on membrane properties concluded a less perturbing effect while experimental assays even point to a fluidizing effect, in line with the general action of related phytochemicals[23,42,43]. We attribute this apparent discrepancy to the lower concentrations of RES (typically

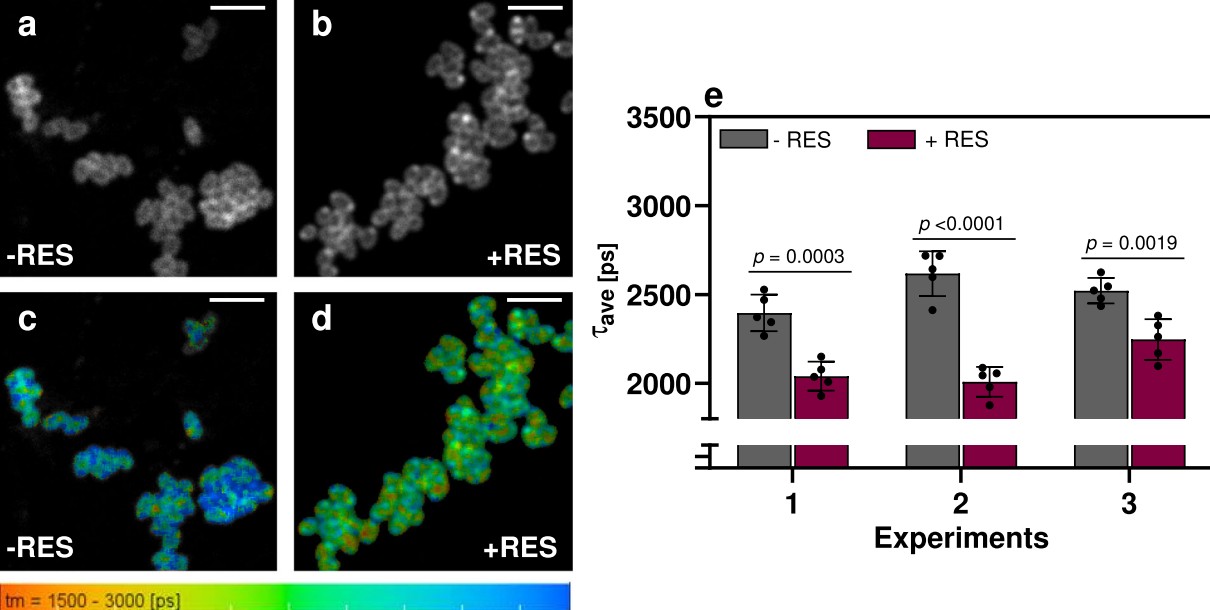

**Fig. 5 | Fluorescence lifetime imaging of Laurdan-stained *C. glutamicum* cells.**
**a**, **b** Fluorescence intensity and **c**, **d** lifetime images of Laurdan-stained *C. gluta-micum* cells (scale bar: 5 μm). Fluorescence lifetime was compared to *C. glutamicum* cells pre-incubated in the absence (**a**, **c**) or presence (**b**, **d**) of RES. **e** Average fluorescence lifetime of *C. glutamicum* cells in presence of RES (magenta) and absence of RES (grey) and standard deviation of five independent preparations from three independent experiments (*n* = 5). Statistical significance was calculated by a two-tailed unpaired Student's *t* test. Source data are provided as a Source data file.

~10 mol%) compared to our study (50 mol%), together with the dif-ference in membrane composition. Most of the previous published results are based on model lipid membranes, whereas our results are obtained with a realistic multi-component mixture representing the plasma membrane of *C. glutamicum*. To shed further light on this issue, we performed additional simulations of RES in a POPC model mem-brane, at 0, 10 and 50 mol%. We obtain area compressibilities that are rather insensitive to RES concentration (Supplementary Table 1), sug-gesting it is the particular lipid composition that is responsible for the observed rigidifying effect of RES in *C. glutamicum*.

**Supplementation of fatty acids counteracts resveratrol effects**
Analysis of the membrane fluidity via Laurdan assays during RES pro-duction in *C. glutamicum*-RES1 in presence of LA and tPAL showed a strong effect of both FAs over the course of time (Fig. 4a). At the beginning of the cultivation (in absence of LA and tPAL) the GP value was identical to the control cultivations. However, after 24 h the GP values for the LA and tPAL cultivations dropped to 0.13 ± 0.01 and 0.18 ± 0.01, respectively, indicating increased fluidity of the *C. gluta-micum*-RES1 membranes. Until the end of the cultivation these values remained almost constant for both cultivations. The impact of LA and tPAL on the *C. glutamicum*-RES1 membrane was also investigated by Laurdan kinetic measurements (Fig. 4b). Initially, the background emission of Laurdan stained *C. glutamicum*-RES1 cells was followed for 10 min. Subsequently, 10 μM LA or tPAL were added to the stained cells. The same amount of EtOH (0.05% (v v⁻¹)) was added to the control approach to exclude interferences in measurements. Subsequently, the fluorescence emission was followed for 60 min. While the GP value of the control cultivation remained constant (0.27 ± 0.01), the GP value dropped immediately upon the addition of LA or tPAL (0.03 ± 0.01 or 0.1 ± 0.01, respectively), indicating an increased membrane fluidity.

Analysis of the FA composition of the *C. glutamicum*-RES1 mem-brane by gas chromatography revealed that neither LA nor tPAL were incorporated into the membrane phospholipids to a notable extent. Within 72 h of cultivation only up to 0.4% and 1.2% of the total mem-brane FAs of *C. glutamicum*-RES1 could be identified as tPAL and LA,

respectively. Therefore, the presence of LA and tPAL as free FAs in the membrane of *C. glutamicum*-RES1 was investigated after LA/tPAL supplementation by using a combination of both thin layer chroma-tography (TLC) and HPLC-MS. Lipid components of the membrane extracts with similar migratory patterns as LA and tPAL standards were seen in both the control cultivation and the cultivations with supple-mented LA and tPAL (Supplementary Fig. 6). The intensity of bands located at this migratory level was found to be significantly higher for both cultivations that had LA and tPAL additives, when compared to the cultivation with no FA supplements. Using HPLC-MS, the presence LA and tPAL as free FAs was confirmed in those cultivations with LA and tPAL supplements, respectively. No detectable LA or tPAL was seen as a free FA in the control cultivation, nor in the reciprocal sup-plement condition, suggesting that supplemented LA and tPAL exist in the membrane as free FAs.

These experimentally observed positive effects of tPAL and LA supplementation on the membrane fluidity of *C. glutamicum* are also supported by MD simulations of *C. glutamicum* model membrane. Here, addition of LA or tPAL to the simulations in similar amounts to RES, increased lipid diffusion rates and membrane fluidity in both cases. In some simulations, FA supplementation even over-compensated the negative RES effects resulting in model membranes with increased lipid diffusion and membrane fluidity (Fig. 4c–e, Sup-plementary Fig. 5, and Supplementary Movies 3 and 4). Furthermore, LA or tPAL also restored the lateral organization of the membrane lipids suppressing the formation of the previously observed RES-induced nanoscale lipid domains.

**Multiple fatty acid supplementations increase product synthesis**
At this stage, both FAs were always added at the beginning of the cultivation. When considering that the biomass increases over the course of cultivation, one could assume that a repetitive addition of LA or tPAL would further increase product formation. Since high LA concentrations have a negative impact on growth and RES production of *C. glutamicum*-RES1, 40 μM LA or 40 μM tPAL were added three times at defined time points (0 h; 24 h; 48 h) over the course of

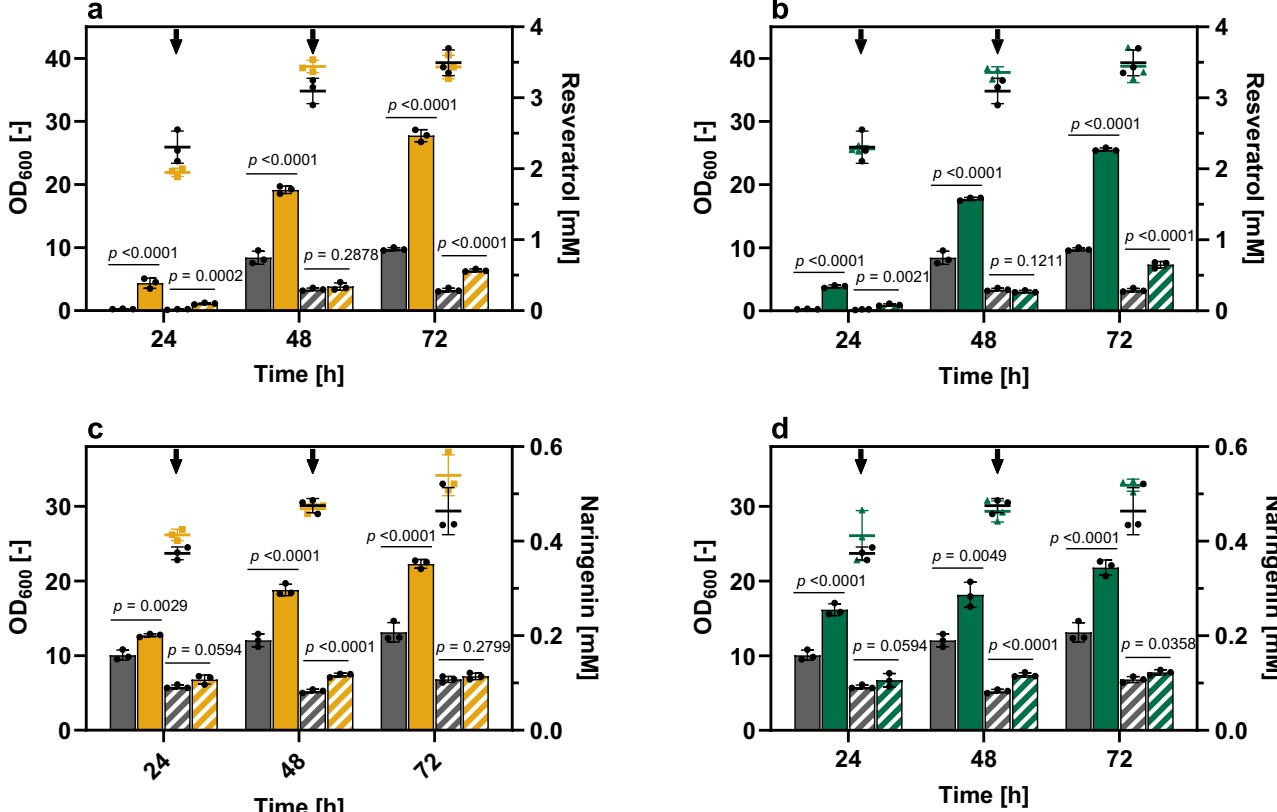

**Fig. 6 | RES and NAR production with multiple supplementations of selected FAs. a**, **b** RES and **c**, **d** NAR production in presence of LA (yellow) and tPAL (green). 40 μM of LA and tPA were added 24 and 48 h after inoculation (indicated by arrows). 0.1 (v v⁻¹) % EtOH was added to the control (grey), since FAs stocks were dissolved in EtOH. RES and NAR were quantified in the culture (cells and supernatant; filled bars)) and in the culture supernatant only (striped bars). Growth in

presence or absence of the FAs was followed over time by determining the optical density of the cultures at 600 nm ($OD_{600}$) (control: black dots, LA: orange squares, tPAL: green triangles). Data represent average values and standard deviation of three biological replicates ($n = 3$). Statistical significance was calculated by a two-tailed unpaired Student's $t$ test. Source data are provided as a Source data file.

cultivation. Compared to the single supplementation of LA or tPAL, a repetitive FA addition did not affect biomass formation of *C. glutamicum*-RES1 (Fig. 6a and b). In contrast, multiple supplementations of LA and tPAL resulted in an increase in RES titer of 17% and 21%, respectively, compared to cultivations with single FA supplementation (EtOH-control: $0.87 \pm 0.02$ mM; LA-pulse: $2.47 \pm 0.07$ mM; tPAL-pulse: $2.28 \pm 0.02$ mM).

Furthermore, it was investigated if the observed positive effect of FA-supplementation on RES production and product export is also transferable to other polyphenols. For this purpose, *C. glutamicum*-NAR, a variant capable of producing the flavonoid naringenin (NAR) was selected[1].

Under standard cultivation conditions, *C. glutamicum*-NAR accumulated in total $0.21 \pm 0.02$ mM NAR in defined CGXII medium containing 4% D-glucose and 5 mM *p*-CA (Fig. 6c and d). Multiple supplementations of 40 μM LA or tPAL at defined time points (0 h, 24 h, 48 h) resulted in a two-fold increase in NAR titer (LA-pulse: $0.45 \pm 0.01$ mM; tPAL-pulse: $0.52 \pm 0.03$ mM). Hence, the positive effect of single or multiple LA/tPAL supplementation(s) on growth and product formation appears not to be limited to stilbenoid-producing *C. glutamicum* variants and can be transferred to other polyphenol-producing cell factories.

Finally, it was evaluated whether supplementation of LA or tPAL has also a positive impact on microbial RES synthesis starting from cheap D-glucose instead of the RES-precursor *p*-CA. Therefore, *C. glutamicum*-RES2 was constructed, which was engineered for the additional heterologous expression of genes for the 3-deoxy-D-arabino-heptulosonate-7-phosphate (DAHP) synthase (AroH) from *E. coli* and

the codon optimized tyrosine ammonia lyase (TAL) originating from *Flavobacterium johnsoniae* in comparison to *C. glutamicum*-RES1. These two additional enzymatic activities enable the overproduction of L-tyrosine and the conversion of L-tyrosine to *p*-CA. Cultivation of this strain variant in the presence of 40 μM LA or tPAL allowed for three-fold increased RES accumulation and export across the membrane of *C. glutamicum*-RES2 in comparison to suitable controls. These results underline the positive effect of FA supplementation on microbial plant polyphenol synthesis (Supplementary Fig. 7).

## Discussion

Product toxicity is a common challenge during the development of microbial production processes. Therefore, the development of robust microorganisms with high tolerance to toxic products is essential for industrial applications[44]. Often, the overall bioprocess performance is negatively influenced by the barrier function of the cell envelope, in particular when no dedicated exporters for product transport are present. As a result, product diffusion or product transport cannot keep up with the product synthesis rate, leading to intracellular product accumulation with detrimental effects on the cellular metabolism[45].

In recent years, *C. glutamicum* emerged as promising host for the production of plant polyphenols such as stilbenoids and flavonoids[12–14,19,46] and the impact of the stilbenoid RES on *C. glutamicum*-RES1 was investigated during growth studies in defined CGXII medium[19]. In this context, it was shown that RES production was indeed limited by the toxic effects of increasing product concentrations accumulating over time. Based on this finding, we investigated

the localization of RES during production in *C. glutamicum*-RES1 as part of this study and could show that RES predominantly accumulates in the cytoplasm and in the fraction of insoluble cell components (membrane and cell wall). In addition, Laurdan fluorescence measurements and FLIM indicated an increased membrane rigidity in the presence of RES.

In contrast to other Gram-positive bacteria, all members of the suborder *Corynebacterineae* share an atypical cell envelope architecture, which is characterized by the presence of a mycomembrane as additional membrane outside of the peptidoglycan layer[47]. The major components of the mycomembrane of *Corynebacterium* are mycolic acids, 2-alkyl- and 3-hydroxy- long chain FAs[48]. This mycolic acid residues are covalently linked to arabinogalactan, which in turn is linked to peptidoglycan forming the mAGP-complex[38]. Depending on the structure of the mycolic acids, the mycomembrane can act as permeability barrier similar to the outer membrane (OM) of Gram negative bacteria. However, recent studies show that the mycomembrane of *C. glutamicum* is highly fluid and does not represent a diffusion barrier compared to the plasma membrane[49,50]. The *C. glutamicum* membrane comprises a largely negatively charged inner membrane (IM) containing phosphatidylglycerol (PG), phosphatidylinositol (PI), as well as cardiolipin and special sugar lipids[37]. Furthermore, the major glycerol lipids of *C. glutamicum* contain predominantly 18:1, 16:0, 18:0, and 16:1 FAs[27]. Previous MD simulation studies investigated the interaction and location of RES within different model membranes (mainly POPC or DPPC)[23,51]. In this context, the hydroxyl groups of RES have been shown to interact with the head groups of phospholipids resulting in localization within the polar head group of membrane phospholipids[52] However, the IM of *C. glutamicum* does not contain phosphatidylcholine (PC) or phosphatidylethanolamine (PE) as lipid head groups[47]. Therefore, a more realistic *C. glutamicum* model membrane was designed to provide a molecular-level interpretation of the effects of RES on the membrane. Based on this, a *C. glutamicum* model membrane containing 50% POPG, 25% DPPI, 12.5% (PO)MGDG and 12.5% PODG was used. MD simulations revealed that the observed accumulation of RES in membrane and cell wall of *C. glutamicum* has drastic effects on lateral membrane organization with a concomitant increase of rigidity and decrease in lipid mobility.

Supplementation with permeabilizing agents can be attempted to increase product transport across the cell envelope. In previous studies, the addition of permeabilizing agents such as TritonX-100 or Tween 80 was shown to improve substrate uptake[53,54]. In the case of *C. glutamicum*, supplementation of ethambutol or Tween 40 has been described to improve product efflux in the context of L-glutamate production[55,56]. Later studies demonstrated that addition of these detergents increases the tension of the *C. glutamicum* membrane, which in turn activates the mechanosensitive channel MscCG responsible for L-glutamate efflux[57–59]. However, addition of such detergents can cause severe membrane damage or even complete cell lysis with negative effects on the microbial metabolism and overall productivity[60]. Additionally, the use of permeabilizing agents can complicate any downstream process with detrimental effects on the product yield[45]. Consequently, the need for alternative methods for an improved product transport across the cell membrane is of high interest.

In this study, we developed a FA supplementation strategy with the aim to decrease the cytotoxic RES-effects and to increase RES export across the cell envelope. A detailed screening identified the unsaturated FAs LA and tPAL as suitable FA supplements to fulfill these requirements. Our data show that LA and tPAL are only incorporated into phospholipids of *C. glutamicum* at low quantities, but remain in the cell envelope as free FAs. This is in contrast to studies showing that other bacteria such as *E. coli*[61], *Streptococcus agalactiae*[62], or *Staphylococcus aureus*[63] are able to incorporate exogenous FAs into the cell membrane[64]. Our experiments and MD simulations showed that both

FAs counteract the negative effects of RES; while RES increases membrane rigidity and lowers lipid diffusivity, supplementation of LA or tPAL restores the original cell envelope properties. In situ extraction of RES by the FAs in the culture supernatant can be excluded due to the FAs concentration used being below the critical micelle concentration of 60–100 μM[65].

With regard to growth inhibitory effects of supplemented FAs, tPAL would be more suitable for scale-up applications, as high tPAL concentrations had no negative effect on the growth of *C. glutamicum*. However, the observed growth-inhibitory effect of high LA concentrations can be circumvented by multiple supplementations of low LA concentrations over the course of cultivation. It is worth noting that LA has a significant advantage in terms of pricing as it is ten times more economical than tPAL.

In addition, we could demonstrate that the observed positive effects of FA supplementation during polyphenol synthesis with *C. glutamicum* are not limited to the microbial synthesis of stilbenoids such as RES, but can be transferred to the production of flavonoids such as NAR. We believe that this cost-effective approach is not limited to polyphenol production with *C. glutamicum* and will prove to be beneficial for the production of a broad range of valuable small and hydrophobic molecules with various microbial cell factories.

## Methods

### Bacterial strain, media, and cultivation conditions

In this study, prophage-free *C. glutamicum* ATCC 13032 variant *C. glutamicum* MB001(DE3) DelAro[4] *4cl* C7 mufasO$_{BCD1}$ was used as chassis strain for the microbial production of RES1, RES2 and NAR[19]. This *C. glutamicum* variant is devoid of an aromatics catabolism and characterized by a high intracellular malonyl-CoA availability, which makes this strain also suitable for the synthesis of other biotechnologically interesting stilbenoids as well as flavonoids or other polyphenols[12–14,66]. For RES-production, this strain was transformed with plasmid pMKEx2_*sts$_{AhCg}$*_*4cl$_{PcCg}$* yielding the strain *C. glutamicum*-RES1. The plasmid enables the episomal expression of the 4-coumarate: CoA ligase (4CL) gene from parsley (*Petroselinum crispum*) and the *sts$_{AhCg}$* gene encoding the stilbene synthase from peanut (*Arachis hypogea*)[46]. Both genes were codon-optimized for heterologous expression in *C. glutamicum* and are organized as bicistronic operon under control of the T7-promoter. *C. glutamicum*-RES2 was transformed with the second plasmid pEKEx3_aroH$_{Ec}$_tal$_{FjCg}$ to facilitate the expression of the native L-tryptophan sensitive 3-deoxy-D-arabino-heptulosonate-7-phosphate (DAHP) synthase (AroH) from *E. coli* and the codon optimized tyrosine ammonia lyase (TAL) from *Flavobacterium johnsoniae*. *C. glutamicum*-NAR is derived from the same chassis strain but harbors the production plasmid pMKEx2_*chs$_{PhCg}$*_*chi$_{PhCg}$* for the synthesis of the flavonoid NAR instead[42]. The plasmid enables the expression of the codon-optimized genes encoding chalcone synthase (CHS) and chalcone isomerase (CHI) from *Petunia x hybrid*[13,46]. Again, both genes were codon-optimized for heterologous expression in *C. glutamicum* and are organized as bicistronic operon under control of the T7-promoter.

*C. glutamicum* was routinely cultivated aerobically at 30 °C in BHI medium (Difco Laboratories, Detroit, MI, USA) or defined CGXII medium[67], containing 4% (w v$^{-1}$) D-glucose as sole carbon and energy source. *C. glutamicum* was grown in test tubes with 5 mL BHI medium for 6 h in a rotary shaker at 170 rpm (first preculture). The whole culture was subsequently used to inoculate 50 mL defined CGXII medium containing 4% (w v$^{-1}$) D-glucose in 500 mL baffled Erlenmeyer flasks and cultivated overnight on a rotary shaker at 130 rpm (second preculture). Main RES production cultures in shake flasks were routinely inoculated to an OD$_{600}$ of 5 in defined CGXII medium with 4% (w v$^{-1}$) D-glucose, 5 mM *p*-CA as RES precursor and 25 μg mL$^{-1}$ kanamycin. Heterologous gene expression was induced 1 h after inoculation using 1 mM of isopropyl $\beta$-D-1-thiogalactopyranoside (IPTG). Bacterial growth was

followed by measuring the optical density at 600 nm or cell dry weight (CDW). CDW was determined gravimetrically. 2 mL culture broth were collected in a weighted reaction tube and centrifuged in a tabletop centrifuge at maximum speed for 10 min. Cells were washed in 0.9% (w v$^{-1}$) NaCl and centrifuged to discard the supernatant. Cell pellets were dried at 60 °C for 24 h followed by an incubation in a desiccator for another 24 h to remove possible humidity residues. Dried cell pellets were weighed afterwards.

FA supplements (Remembrane Srl, Imola, Italy) were supplied as EtOH stock solutions and supplemented to *C. glutamicum* cultivations to a final concentration of 40 µM if not stated otherwise. Shake flask cultivations were conducted as biological triplicates for 72 h. Samples were taken at defined time points for the quantification of the cell density as well as product concentrations and stored at −20 °C until further processing. Growth experiments with FAs as sole carbon and energy source were conducted in a BioLector microbioreactor using 48 well microtiter flowerplates (Beckman Coulter, Baesweiler, Germany) at 30 °C, 1000 rpm and 85% humidity containing 900 µL defined CGXII medium with 40 µM of the respective FA. Cells were inoculated to an OD$_{600}$ of 5. Backscattered light intensity (620 nm, gain 20) was followed online and used to follow cell growth.

### Preparation of cellular fractions
After cultivation of *C. glutamicum* in defined CGXII medium containing 4% (w v$^{-1}$) D-glucose and 5 mM *p*-CA for 72 h, the cell dry weight and OD$_{600}$ of the cultures were quantified. *C. glutamicum*-RES1 cells were separated from the culture medium by centrifugation at 6000 × *g*, 4 °C for 20 min. Cell pellets were washed twice with 100 mM of cold potassium phosphate buffer, which was adjusted to pH 7. The volume of the cell pellets was determined in scaled conical test tubes and cell pellets were suspended in a defined volume of 100 mM cold potassium phosphate buffer. 1 mL of washed cell suspensions and supernatants were stored at −20 °C until further processing.

To quantify RES in the cytoplasm and in insoluble components (membrane and cell wall components) of *C. glutamicum*-RES1 cells, suspensions of washed cells were disrupted individually by six passages through a French Press at 18,000 psi on ice (G. Heinemann, Schwäbisch Gmünd, Germany). The resulting cell extracts were subjected to ultracentrifugation at 50,000 × *g*, 4 °C for 60 min. Subsequently, the disrupted cell pellets were separated from the supernatant and then washed twice with 50 mL of cold 100 mM potassium-phosphate buffer and centrifuged again for 20 min at 6000 × *g* to determine the pellet volume in scaled conical tubes. Obtained supernatants were used to quantify the RES concentration in the cytoplasm. After establishing the correlation between OD$_{600}$ and cell dry weight for *C. glutamicum*-RES1 (Supplementary Fig. 8), cytoplasmic volumes of *C. glutamicum*-RES1 were determined[68]. The content of the cytoplasm and the suspension of the disrupted cells were stored at −20 °C until further processing.

### Polyphenol quantification
For quantification of RES and NAR in the culture broth, 0.5 mL culture were mixed with 1 mL ethyl acetate and shaken vigorously (1400 rpm, 10 min, 20 °C) in a thermomixer (Eppendorf, Hamburg, Germany). Suspensions were centrifuged for seven min at 16,000 × *g* ethyl acetate layer (400 µL) transferred to an organic solvent resistant deep-well plate (Eppendorf, Hamburg, Germany). Ethyl acetate was evaporated overnight, and dried extracts were subsequently resuspended in the same volume of acetonitrile and directly used for HPLC-MS analysis. RES was quantified in the culture and supernatant by HPLC-MS using an ultra-high-performance LC (uHPLC) 1290 Infinity System coupled to a 6130 Quadrupole HPLC-MS System (Agilent Technologies, Waldbronn, Germany). LC separation was carried out using a Kinetex 1.7 µm C18 100 Å pore size column (2.1 × 50 mm; Phenomenex, Torrance, CA,

USA) at 50 °C. As eluent 0.1% acetic acid (solvent A) and acetonitrile supplemented with 0.1% acetic acid (solvent B) were applied as the mobile phases at a flow rate of 0.5 mL min$^{-1}$. A gradient was used, where the amount of solvent B was increased stepwise: min 0 to 6: 10% to 30%, min 6 to 7: 30% to 50%, min 7 to 8: 50% to 100% and min 8 to 8.5: 100% to 10%. The mass spectrometer was operated in the negative electrospray ionization (ESI) mode. Data acquisition was performed in selected ion monitoring (SIM) mode. Authentic RES and NAR standards were purchased from Sigma-Aldrich (Schnelldorf, Germany). Area values for [M-H]$^-$ mass signals were linear up to metabolite concentrations of at least 250 mg L$^{-1}$. Benzoic acid (final concentration 100 mg L$^{-1}$) was used as internal standard. Calibration curves were calculated based on analyte/internal standard ratios for the obtained area values. All values are the arithmetic mean of three biological replicates and error bars demonstrate the standard error of the mean. Statistical significance was estimated by a two-tailed *t* test (heteroscedastic, $p \leq 0.05$).

### Laurdan assays
Membrane fluidity of treated and untreated samples was assessed by Laurdan assays. For these assays, culture samples were taken at defined time points during cultivation and diluted to an OD$_{600}$ of 1. 1 mL of these cell suspensions were transferred to 2 mL reaction tubes and supplemented with 10 µM Laurdan (6-dodecanoyl-2-dimethylaminonaphtalene; Sigma-Aldrich, Schnelldorf, Germany) from a 1 mM Laurdan stock solution solved in DMSO. Cells were incubated with Laurdan for 20 min at 30 °C in a thermomixer, which was covered with aluminum foil to prevent any exposure to light. Subsequently, cells were centrifuged for 1 min at 16,000 × *g* in a benchtop centrifuge. Cells were washed four times with 2 mL pre-warmed 1x potassium phosphate buffer (PBS) solution. Supernatants were carefully removed by pipetting and after the last washing step, cells were resuspended to an OD$_{600}$ of 0.5 in 1× PBS. Half of the suspensions were transferred to new reaction tubes and centrifuged for 10 min at 16,000 × *g*. Subsequently, supernatants were harvested and used as Laurdan background fluorescence (background of buffer and cell unassociated Laurdan dye) in successive measurements. 150 µL of stained cells and Laurdan background were instantly transferred to pre-warmed black, flat bottom 96- well microtiter plates. Laurdan fluorescence reader (excitation: 350 nm, emission: 420–460 nm and 490–520 nm) was immediately determined using a filter-based Infinite 200 PRO plate (Tecan, Männedorf, Switzerland).

For kinetic measurements, 150 µL Laurdan stained or untreated cells (in both cases adjusted to an OD$_{600}$ of 0.5) as well as Laurdan background were transferred into a pre-warmed black, flat bottom 96- well microtiter plates. Fluorescence of untreated cells was measured in a fluorescence plate reader in a 1 min time interval over 10 min (pre-treatment baseline). 0.5 µM of LA or tPAL as well as 0.5 mM RES were added to Laurdan stained cells. Since the FAs were stored in EtOH but RES in DMSO, the same amounts of EtOH or DMSO respectively, were added to the untreated stained cells serving as control. Fluorescence measurements were continued for 60 min.

Data were analyzed by subtracting background values from treated and untreated cell suspension values. The Laurdan generalized polarization (GP) was calculated as followed[29]:

$$GP = \frac{I_{435} - I_{500}}{I_{435} + I_{500}} \quad (1)$$

### Laser scanning fluorescence microscopy
For microscopy, a small aliquot of the Laurdan-stained bacteria was gently pipetted onto a glass slides for microscopy, which had been coated with an agar-pad. Fluorescence Lifetime Imaging with pulsed, two-photon excitation was performed on a laser scanning fluorescence

microscope (LSM880, Zeiss, Jena, Germany) equipped with a ×20 water immersion objective (NA 1.0, WD 2.1 mm; Zeiss, Jena). Laurdan was excited with 780 nm (120 fs pulses with 80 MHz repetition frequency) and its fluorescence was observed after passing through a broad band-pass filter centered at 445 nm (FWHM 90 nm; Omega Optical, Brattleboro, VT, USA) covering the blue part of the Laurdan emission spectrum, whose fluorescence decay is sensitive to the membrane fluidity. Fluorescence photons were detected with a GaAsP hybrid photodetector (HPM-100-40, Becker & Hickl, Berlin, Germany). TCSPC electronics (SPC-152; Becker & Hickl, Berlin, Germany) and acquisition software were used for FLIM[69,70]. Fluorescence lifetime images were generated using SPCImage 8.4 (Becker & Hickl, Berlin, Germany). Fluorescence decays were fitted with bi-exponential functions[71–73] and the average fluorescence lifetime ($\tau_{ave}$):

$$\tau_{ave} = (a_1 \cdot \tau_1 + a_2 \cdot \tau_2)/(a_1 + a_2) \qquad (2)$$

was used to represent the Laurdan fluorescence decays satisfactorily ($\tau_i$ = lifetime of the $i$th exponential component; $a_i$ = respective amplitude).

### Identification and quantification of fatty acids
*C. glutamicum*-RES1 biomass pellets, pre-washed in cold PBS, were resuspended in lysis buffer (50 mM Tris, 50 mM NaCl, 250 mM sucrose) and were disrupted using a French Press at 18,000 psi on ice. The lysate was then centrifuged to pellet unbroken cells and cell debris (800 × $g$, 10 min, 4 °C). The supernatant was subsequently centrifuged (100,000 × $g$, 30 min, 4 °C), and the resulting pellets were used for membrane lipid extraction.

For TLC and HPLC-MS analysis, the membrane lipids were extracted using a modified Bligh and Dyer method[74]. The extracted lipids were stored under nitrogen gas prior to analysis to prevent oxidation of the unsaturated bonds. In order to standardize loading onto TLC plates, the total lipids were roughly quantified using an ammonium ferrothiocyanate assay[75]. Approximately 20 μg of extracted lipid and 60 μmol LA and tPAL standards were loaded onto aluminum-backed silica gel TLC plates (Sigma-Aldrich, Schnelldorf, Germany). The lipid components were separated using a cyclohexane-ethyl acetate (3:2) mobile phase. Plates were allowed to dry in air before being dipped into a 10% CuSO₄ solution. Following this, the plates were charred at 150 °C for 10 min. Band intensity was calculated using ImageJ v1.53m (Bethesda, MD, USA).

HPLC-MS was performed on an Acquity I-class LC systems (Waters Corporation, Milford, MA, USA) coupled to a Select Series Cyclic IMS system (Waters), running under the MassLynx Software systems (Waters Corporation, Milford, MA, USA). Chromatography was performed on a 2.1 × 150 mm, 2.6 μm Accucore C30 column (Thermo Fisher Scientific Inc., Waltham, MA, USA). All solvents were OptiPure grade and all reagents were HPLC-MS grade. Samples were dissolved in 100 μL chloroform and diluted 1:100 in isopropanol, with 5 μL being loaded on to the column. Elution was performed at 200 μL min⁻¹ with a gradient elution of: 0 min 90% A, 4 min 70% A, 6 min 60% A, 16 min 10% A, 21 min 1% A, 24 min 1% A, 25 min 90% A, 30 min 90% A. Solvent A was 50:50 acetonitrile:water, 10 mM ammonium formate, 0.1% formic acid, and solvent B was 85:10:5 isopropanol:acetonitrile:water, 10 mM ammonium formate, 0.1% formic acid. Mass spectrometry data was collected in MS$^E$ V-mode in the range 100–2000 $m/z$, with 0.5 s acquisitions, low energy collision energy set at 4 V and high energy ramped from 25 to 50 V, using 50 μg mL⁻¹ leucine encephalin as the lock mass. Data was analyzed using Progenesis QI software v2.6 (Waters Corporation, Milford, MA, USA).

### Identification and quantification of membrane lipids
To extract membrane lipids, *C. glutamicum*-RES1 was cultivated in defined CGXII medium for 72 h. Cells were harvested at defined time-points and separated from the culture supernatant (6000 × $g$, 15 min, 4 °C). The cell pellets were washed three times with cold PBS and subsequently disrupted individually by six passages through a French Press at 18,000 psi on ice (G. Heinemann, Schwäbisch Gmünd, Germany). The resulting cell extracts were centrifuged (6000 × $g$, 30 min, 4 °C) and pellets were used for membrane lipid extraction. Extracted samples were stored at −20 °C until analytical measurements.

Membrane lipids of *C. glutamicum*-RES1 were extracted with CHCl₃/MeOH (2:1 v v⁻¹) and then incubated with 0.5 M KOH in methanol for 10 min at room temperature, thus *trans*-esterifying FAs linked by ester bonds to methanol to form fatty acid methyl esters (FAMEs). FAMEs were extracted with *n*-hexane and separated by GC using an Agilent 7820 A GC System (Agilent Technologies, Waldbronn, Germany) fitted with a 30 m × 0.25 mm DB23 capillary column, film thickness 0.25 μm, and a flame ionization detector (FID). Helium was used as a carrier gas at 1.21 mL min⁻¹ and the spilt injector was used with a split ratio of 10:1. Injector temperature was 250 °C and detector temperature was set to 300 °C. The column oven temperature was maintained at 50 °C for 1 min after sample injection and programmed for the following temperature gradient: 25 °C min⁻¹ from 50 °C to 175 °C, 4 °C min⁻¹ from 175 °C to 230 °C and holding at 230 °C for 5 min. The separation was recorded with G6714AA SW EZChrom Elite Compact (Agilent Technologies, Waldbronn, Germany). FAMEs were identified by comparison with standards purchased from NuCheckPrep Inc. (Elysian, MN, USA). FAMEs are expressed in weight %, based upon the contribution of the peak area of each FAME in the chromatogram. To take into account the different signal of the detector for different molecules, a correction factor was applied to the experimental data from the integration of the chromatograms. The total of the peaks analyzed for each chromatographic run was 100.

### MD simulation setup and composition
The coarse grained Martini model (version 2)[76] was used to describe molecular interactions in MD simulations. The following composition was chosen to model the plasma membrane of *C. glutamicum* based on membrane lipid studies of Nakayama et al.[37] and Bansal-Mutalik et al.[38]: 50% POPG, 25% DPPI, 12.5% PODG and 12.5% (PO)MGDG. The initial conditions of the simulated lipid bilayers were generated using the tools Insane and Martinate[77] to yield lateral dimensions of 30 × 30 nm with the bilayer repeat distance of 20 nm. The final systems consisted of 3200 lipids and 206,875 water beads (one CG water bead representing 4 water molecules). The added concentrations of each of the used supplements (RES or FAs) was 1:1 supplement:POPG. The lipid and FA parameters were taken from Wassenaar et al.[77], the RES parameters from Ingólfsson et al.[23]. The systems were equilibrated for at least 10 ns prior to production simulations. The simulation temperature was coupled to a v-rescale thermostat[78] at room temperature of 293 K, separately for the lipids and solvent. A Parrinello-Rahman barostat[79] was used for pressure coupling at 1 bar with a coupling constant of 12 ps independently for the membrane plane and its normal. Standard time step of 20 fs was used for all simulations and the trajectory was recorded every 1 ns. For production, we have simulated all systems for 1 μs to ensure convergence; the last 0.5 μs was used for analysis. Simulations were run using GROMACS[80] simulation package v2019.3 in a mixed precision compilation with GPU support. The planar radial distribution functions, diffusion coefficients and compressibility coefficients were calculated using built-in tools in the GROMACS[80] package. The calculation of the diffusion coefficient from mean square displacement was done using the Einstein relation. Standard formula for area fluctuation[81] was used to calculate the compressibility coefficients. The error estimates of both coefficients were calculated using block averaging by halving the data sets in the temporal domain. The RDFs were

computed between the POPG headgroups (using the phosphate bead) in the plane of the membrane. Error bars were computed as standard deviation considering both leaflets as independent measurements. Snapshots from the simulations, as well as movies, were prepared using VMD v1.9.3[82].

## Reporting summary

Further information on research design is available in the Nature Portfolio Reporting Summary linked to this article.

## Data availability

Data supporting the findings of this work are available within the article and its Supplementary Information file. A reporting summary for this article is available as a Supplementary Information file. Source data are provided with this paper.

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

## Acknowledgements
This work was funded by the German Federal Ministry of Education and Research (BMBF, Grant. No. 031B0621, J. Marienhagen) and BBSRC award BB/R02152X/1 (A.D.G.) as part of the project "MeMBrane - MEmbrane Modulation for BiopRocess enhANcEment", which is embedded in the ERA CoBioTech action of the ERA-NET Cofund under the European Union's Horizon 2020 research and innovation program.

## Author contributions
A.T. and J. Marienhagen conceived and designed the study. A.T. determined RES concentrations in different cellular fractions and performed all Laurdan assays. A.T., K.K., and K.M.E. performed cultivations with FS supplementations in shake flasks. J.L. and A.D.G. performed TLC experiments and A.R.P. performed HPLC-MS analysis of the TLC samples. P.P. and A.C. supplied the FA supplements and analyzed membrane FAs using GC. T.G. conducted all fluorescence microscopy measurements and collected the FLIM data. J. Melcr, C.B. and S.J.M. performed all computational modeling experiments. A.T. and J. Marienhagen wrote the manuscript with input from all authors.

## Funding

## Competing interests
The authors declare no competing interests.
