## [Peer Review File · Nature Communications]

Membrane manipulation by free fatty acids improves microbial plant polyphenol synthesisReviewers' Comments:

Reviewer #1:

Remarks to the Author:

In this manuscript the authors demonstrate how resveratrol production by *Corynebacterium glutamicum* can be boosted by addition of free fatty acids (FA), where the free fatty acids balance and/or compensate for change in membrane properties caused by the buildup of resveratrol, allowing for higher levels of production. Overall, I find the article well written, and the appropriate set of experiments and simulations done to highlight the change in flavonoid production, narrowing down to the likely case of membrane disruption, and the compensatory change in membrane properties by FA addition. That being said a number of questions arose while reviewing the manuscript which I hope the authors can comment on, as well as some minor comments listed below.

RES clearly partitions in or on membranes and changes membrane properties. Here the authors call it "The membrane rigidifying effect of RES" and indeed both experiments and simulations in this manuscript point to less fluid membranes. But other studies on RES, e.g.

<https://pubs.acs.org/doi/10.1021/cb500086e>

<https://pubs.acs.org/doi/10.1021/acs.jcim.2c00372>

<https://dx.doi.org/10.1021/acs.jmedchem.0c00958>

seem to indicate a membrane softening effect of RES, at least simulation work with simple PC membranes and at lower RES concentrations make the membrane easier to deform, better fitting softening then stiffening/rigidifying effects. Would be good if the authors could comment on that, so future readers won't have to speculate, and if they think this is due to the higher concentration of RES and/or bilayer composition (potential POPG clustering)?

For the nanoclustering of POPG with RES e.g. "The addition of RES brings the most abundant POPG lipids in closer proximity and increases their segregation with the other lipids, causing the formation of transient nanoscale lipid domains (Supplementary Fig.4, Supplementary Movie1+2).", the POPG-POPG RDFs in figure 4d look clear but in the images in Fig.S4 or movies 1 and 2 for me it is very hard to see any nanoclustering. Is this nanoclustering one or a few RES molecules bringing together a handful of POPG, or is this POPG lipids mixing slightly less with the other three lipid types? Maybe some additional analysis would make this easier to interpret.

There is one question that I know might not be possible to answer, but I would love the authors to comment on, pertains to the choices made for FA and RES concentrations in the manuscript. Concentrations can be quite tricky to match between different experimental setups and are in very different frame of reference in the simulations. For the growth experiments 40 μ M FA was found to be best, producing something \sim 1.5 mM RES. But in the Laurdan measurements RES was similar (0.5 mM) but why was there only 0.5 μ M FA used? And are the same concentrations used in Fig.5? For the simulations was a 50mol% RES/FA selected just because it was very high, or is it possible to estimated/match the experiments?

As free fatty acids at pH 7 would likely be in a mixture of protonation states, and the protonation state is likely to influence their bilayer modifying effect e.g. <https://pubmed.ncbi.nlm.nih.gov/23805753>, was that considered in the simulations here?

Minor comments:

- L61, Reference 9, "Wang, E. & Jeffery 2017", looks out of place and not related to polyphenol production.

- L171, Fig. 3c y-axes labels 18:2n9 should be 18:2n6.

- L165, Fig. 5 caption scale bar 5 μM likely should be μm
- L224, Fig. 4. In panel b LA tPAL addition shows a response with 2.5 min but in panel a there is no effect at 0, this was very confusing until I read the associated main text "At the beginning of the cultivation (in absence of LA and tPAL)", to avoid similar confusion by others maybe some tweak of the figure would be in order, e.g. add line or arrow (to indication FA addition) and/or append caption text to would avoid confusion.
- L278, "overall neutral polar lipids^{33–36}". Some reference mix up e.g. reference 35 is the GROMACS MD package reference.
- L278, "MD simulations demonstrated that RES has a high affinity to the surface of lipid bilayers of the *C. glutamicum* model membrane, which agrees well with previous results from model neutral lipid bilayers (Fig. 4c-d, Supplementary Fig.4)^{23,24,37–39}". Fig. 4c or d do not show RAS membrane affinity and Fig. S4 kind of shows that, i.e. lot of RES in the membrane, but as the water phase is cut off that is not highlighted. RES has a very high water to lipid partitioning so this is not surprising but for the "MD simulations demonstrated that RES has a high affinity to the surface of lipid bilayers" statement maybe something like a density plot would be clearer?
- L597, "Identification and quantification of membrane lipids". Was there any quantification of membrane lipids reported?
- L640, "Error bars were computed from two independent membrane leaflets.". Does that mean range was used?
- The videos that come with the reviewer material were not labeled 1 to 4 name e.g. 405795_0_video_7205006_rncd25 and the quality was rather low, but the download links in the SI did work. For video 4 – without LA for clarity, I think that video also either is not showing or dose not have any RES?

Reviewer #2:

Remarks to the Author:

This paper by Tharmasothirajan et al. proved the toxicity of polyphenol synthesis on the cell membrane and explored how adding free fatty acids can attenuate such toxicity and improve the production of polyphenols. In brief, using resveratrol (RES) as a representative polyphenol, the authors first analyzed its distribution and found out most RES is located in cell pellet with more fraction in cell membrane compared to cytoplasm. The authors then hypothesized that such a high concentration of RES would cause toxicity and inspired by previous studies, they attempted to add different free fatty acids to reduce the toxicity. Among the 12 tested fatty acids, most of them could optimize cell growth and the supplementation of palmitelaidic acid and linoleic acid most effectively increased the RES yield. Subsequently, the mechanism was elucidated by analyzing and modeling the structure and properties of the cell membrane. RES rigidifies the cell membrane and decreases its fluidity, and the incorporation of palmitelaidic acid and linoleic acid into the cell membrane increases its fluidity to counteract the effects of RES. Overall, this is a well-conducted study demonstrating the capability of free fatty acid supplements to relieve the toxicity of polyphenols for higher production. However, I do not think it is novel or influential enough to be published on Nature Communications. As the authors mentioned, the influence of free fatty acids on the cell membrane and tolerance resistance has been explored in many previous studies [1]. In addition, the superiority of this approach was not fully demonstrated. It will be more convincing if the authors can use this approach to optimize previous overproducers and achieve higher titers than previous studies.

The following comments hopefully would help the authors to further improve this paper.

1. Line 118, the authors might want to provide the data of cell growth under different concentrations of RES to fully prove the cytotoxicity of RES.
2. Line 127-131, the related reference should be provided to support that supplementation of FAs can positively influence the membrane composition of some bacteria for enhanced antibiotic resistance.
3. Appropriate references were absent, for example, line 128 and line 246.
4. Line 153-155, "In all cases, the growth behavior was similar to the EtOH control confirming that *C. glutamicum*-RES cannot metabolize any of the supplemented compounds", why did the addition of unsaturated trans-FAs increase the biomass formation by 14 %? Please provide the perspectives.
5. Based on these results, 40 μ M LA or 40 μ M tPAL appear to be sufficient for increased RES production using *C. glutamicum*-RES. To increase the resveratrol titer, LA or tPAL was added repetitively. Whether the resveratrol titer will further increase when FA and tPAL were added simultaneously and repetitively?
6. Although the free fatty acids supplied in this study had been proven not involved in cell metabolism, is there any possibility that the added fatty acid were able to extract the products, thereby alleviating the cytotoxicity of the products?
7. The authors talked a lot about how resveratrol and fatty acid alter cell membrane in detail. However, more explanations about how the changed cell membrane properties facilitate polyphenols need to be provided. For example, fatty acid increased the membrane fluidity, why the increased fluidity contributed to resveratrol production?
8. If the increased membrane fluidity could increase export across the membrane, more products will go to the supernatant. However, more increased production was not accumulated in the supernatant according to Fig.6. When producing naringenin, it even looks like no increase in the supernatant.
9. To better connect cell membrane properties with resveratrol production, more control groups are needed. For example, adding certain reagents rather than fatty acids to change the cell membrane fluidity and measuring the effect on resveratrol production.
10. Fig.3, for the EtOH control group, not only the biomass formation and the resveratrol titer increased, but also the product titer ratio of total RES vs extracellular RES changed, just similar to many FA groups, may EtOH also have effects on membrane characteristics? If EtOH can act on membrane characteristics, how can EtOH impacts be excluded for the experimental groups? Besides, what's the relationship between Fig 3b and 3c, from my understanding, final product titers were normalized to optical density (OD600) to get Fig. 3c, but why only the control group in Fig 3c shows evident differences in the ratio of total RES vs extracellular RES compared to that of 3b?
11. Page 8, "In all cases the growth behavior was similar to the EtOH control confirming that *C. glutamicum*-RES cannot metabolize any of the supplemented compounds." And in page 20, the paper goes that "Our data show that LA and tPAL are only incorporated into phospholipids of *C. glutamicum* at low quantities, but remain in the cell envelope as free FAs." So the supernatant FAs and total FAs titers are suggested to be detected and added, this will help support these two conclusions.
12. There are many grammar issues in this paper. As an example, in line 63, do you mean elucidation by "elimination"? I also do not think "here" is appropriate. Lines 83 and 201 also have grammar issues. The authors should revise the paper carefully.

Reference

1. Sandoval N R, Papoutsakis E T. Engineering membrane and cell-wall programs for tolerance to toxic chemicals: beyond solo genes[J]. *Current opinion in microbiology*, 2016, 33: 56-66.

Reviewer #3:

Remarks to the Author:

This manuscript describes production of resveratrol (RES) by engineered *Corynebacterium glutamicum*. The authors showed that supplementation of palmitelaidic acid (tPAL) and linoleic acid (LA) improved production and extracellular export of RES. By supplementation of 40 μ M LA, 2.47 mM RES was produced from 5 mM p-cumaric acid, which was 2.8-fold higher than that of the control. The

authors also showed that supplementation of tPAL and LA increased and restored the membrane fluidity rigidified by RES synthesis, although the membrane composition was not altered by supplementation of tPAL and LA. In the aspect of basic research, it is interesting that supplementation of the fatty acids improved production and export of RES, but no mechanism was described in this manuscript. Besides, it is already known that the supplementation of fatty acids influences the cellular physiology by changing the membrane composition as stated in Introduction (P5 L86). In the aspect of applied research, it is important to produce industrially important chemicals from inexpensive sugar-based feedstocks is important as stated in Introduction (P3 L58). However, in this manuscript, RES was produced from p-cumaric acid. Besides, the production titer and yield was not prominent compared to the previous studies on RES production (PMID: 30798357). Overall, the reviewer has to conclude that this manuscript does not reach the quality of the journal.

Major comment

P16 L323 How do free LA and tPAL increase the membrane fluidity without changing the membrane composition? We need to know the mechanism.

P17 L344 It is high titer of RES by LA supplementation (2.47 mM (564 mg/L)) for *C. glutamicum* as a host strain, but there is a paper reporting production of higher titer of RES, 1.4 g/L from p-cumaric acid and 2.3 g/L from p-cumaric acid and cerulenin with high yield using *E. coli* (PMID: 21441338). What is the advantage of using *C. glutamicum* for RES production?

P20 L425 Is there a correlation between the effect of RES on decreasing the membrane fluidity and cytotoxicity? Please explain.

P20 L433 How do LA and tPAL in cell envelope counteract the cytotoxic effect of RES in cytoplasm without incorporated in the cell membrane? Please explain.

P20 L434 If the supplementation of LA and tPA just restore the original cell envelop properties, how does it improve resveratrol production and export? Please explain.

Minor comment

P8 L160 In Fig. 3b, it looks like there is clear difference between extracellular and total RES concentrations on supplementation of palmitelaidic, linoleacdic, and linoleic acids.

P8 L165 No data for LA supplementation in Fig. 3C.

P10 L193 In supplementary Fig. 2, 40 μM is the best concentration for RES production? How about 20 or 30 μM ?

Point- by point response to Reviewers

Membrane manipulation by free fatty acids improves microbial plant polyphenol synthesis

Reviewer #1:

*In this manuscript, the authors demonstrate how resveratrol production by *Corynebacterium glutamicum* can be boosted by addition of free fatty acids (FA), where the free fatty acids balance and/or compensate for change in membrane properties caused by the buildup of resveratrol, allowing for higher levels of production. Overall, I find the article well written, and the appropriate set of experiments and simulations done to highlight the change in flavonoid production, narrowing down to the likely case of membrane disruption, and the compensatory change in membrane properties by FA addition. That being said a number of questions arose while reviewing the manuscript, which I hope the authors can comment on, as well as some minor comments listed below.*

1. RES clearly partitions in or on membranes and changes membrane properties. Here the authors call it “The membrane rigidifying effect of RES” and indeed both experiments and simulations in this manuscript point to less fluid membranes. But other studies on RES, e.g. <https://pubs.acs.org/doi/10.1021/cb500086e>

<https://pubs.acs.org/doi/10.1021/acs.jcim.2c00372>

<https://dx.doi.org/10.1021/acs.jmedchem.0c00958>

seem to indicate a membrane softening effect of RES, at least simulation work with simple PC membranes and at lower RES concentrations make the membrane easier to deform, better fitting softening then stiffening/rigidifying effects. Would be good if the authors could comment on that, so future readers won't have to speculate, and if they think this is due to the higher concentration of RES and/or bilayer composition (potential POPG clustering)?

Answer:

Thank you. The mentioned publications mainly describe the effects of chemicals, including RES, on membrane-active proteins. In terms of softening or rigidifying effects of the compounds, the results from the listed studies do not seem conclusive for RES. For instance, the results in Fig S1 from <https://pubs.acs.org/doi/10.1021/cb500086e> do not point to any effect of RES at the used concentration of 10 mol% (in SI therein). Also, the profiles in Figure

1 therein cannot be interpreted unambiguously either. Similar argument holds for Figure 3 in <https://pubs.acs.org/doi/10.1021/acs.jcim.2c00372> or for Figure 3 in <https://dx.doi.org/10.1021/acs.jmedchem.0c00958>.

Since all above mentioned papers use the same MARTINI force field as used in our work, it is a likely interpretation that the effects observed therein are small due to the used low concentration of RES (10mol%) compared to our study (50mol%).

In the revised manuscript, we now mention these other studies and comment on the apparent concentration-dependent effects:

“Interestingly, previous MD studies probing the effect of RES on membrane properties concluded a less perturbing effect, which we attribute to the lower concentrations of RES (~10 mol%) compared to our study (50 mol%)” (line 318).

2. For the nanoclustering of POPG with RES e.g. “The addition of RES brings the most abundant POPG lipids in closer proximity and increases their segregation with the other lipids, causing the formation of transient nanoscale lipid domains (Supplementary Fig.4, Supplementary Movie1+2).”, the POPG-POPG RDFs in figure 4d look clear but in the images in Fig.S4 or movies 1 and 2 for me it is very hard to see any nanoclustering. Is this nanoclustering one or a few RES molecules bringing together a handful of POPG, or is this POPG lipids mixing slightly less with the other three lipid types? Maybe some additional analysis would make this easier to interpret.

Answer:

Thank you, we are glad to see an agreement that the effect is clearly visible in Figure 4d. Due to the small size and transient nature of the nanoclusters, they are indeed hard to observe in the rendered images or videos. However, small differences arise, when the videos are compared, e.g. the lipid mobility is clearly slowed down in the presence of RES in video S2 compared to the control in video S1, to the membrane containing both RES and LA in Video S3, and in video S4 (where LA is not shown for easier comparison with the other videos).

From our gathered data, we can only point to the conclusion that, based on our simulations, RES causes the lipids to reorganize to some extent as seen in the RDF in Figure 4d, pointing to POPG lipids mixing slightly less with the other lipids. The mechanistic view of "RES molecules bringing together a handful of POPG" is not supported by our data. Given the subtlety of this effect, a more thorough interpretation is difficult to achieve, however. In our view, the most remarkable aspect is that both, the rigidifying effect and the changes in lateral organization, are restored when FAs are supplied.

3. There is one question that I know might not be possible to answer, but I would love the

authors to comment on, pertains to the choices made for FA and RES concentrations in the manuscript. Concentrations can be quite tricky to match between different experimental setups and are in very different frame of reference in the simulations. For the growth experiments 40 μM FA was found to be best, producing something ~ 1.5 mM RES. But in the Laurdan measurements RES was similar (0.5 mM) but why was there only 0.5 μM FA used? And are the same concentrations used in Fig.5? For the simulations was a 50mol% RES/FA selected just because it was very high, or is it possible to estimated/match the experiments?

Answer:

Laurdan kinetic experiments were performed with *C. glutamicum* cells resuspended to an OD_{600} of 0.5. At the end of the cultivation experiments, an OD_{600} of 37-40 were determined at an applied fatty acid concentration of 40 μM . To maintain the ratio of *C. glutamicum* cells to FA during the Laurdan kinetic experiments comparable to that at the end of the cultivation, a concentration of 0.5 μM FA was used at an OD_{600} of 0.5.

Comparing concentrations between simulations and experimental measurement for membrane active compounds like RES is not an easy task. Due to the strong affinity of these compounds for the membrane, the aqueous concentration (as reported for the experiments) is not a direct reporter of the mol% present in the membrane.

In our current simulations in this manuscript, the aqueous concentration of RES is very low; see for instance Figure S4, showing no RES outside of the membrane surface (no membrane component was omitted in Figure S4). However, it did happen during the simulations that with a very low probability a RES molecule detached from the membrane and spent some time in the aqueous environment.

In conclusion, while it is still very hard to compare the concentrations between experiments and simulations, we would like to point out that considering 50 mol% concentration in simulations as "very high" is likely inaccurate as the aqueous concentration in the simulations is, on the contrary, very low.

We added a comment in the revised manuscript: *"A direct comparison of RES concentrations with the experimental setup is difficult, as the aqueous concentration is not very informative on the amount of RES bound to the membrane, given the strong affinity of RES for the latter. During the simulations, RES molecules were observed to detach from the membrane with very low probability, subsequently spending some time in the aqueous environment before rebinding."* (line 297)

4. As free fatty acids at pH 7 would likely be in a mixture of protonation states, and the protonation state is likely to influence their bilayer modifying effect e.g.

<https://pubmed.ncbi.nlm.nih.gov/23805753>, Was that considered in the simulations here?

Answer:

No, this effect was not considered in our manuscript. Since the pKa value for LA is above 9, we have assumed the state of the FAs to be protonated (i.e., neutral) in our study. A CurTiPot analysis (http://www.ig.usp.br/gutz/Curtipot_.html) for LA (pKa 9.24) shows that > 98 % are indeed protonated at pH 7. For tPAL, pKa-value has not been experimentally determined, but the very similar *trans*-elaidic acid (18:1, pKa = 9.95) is also almost fully protonated at pH 7. Hence, it appears safe to assume the effect of the small deprotonated population of fatty acids at pH 7 as negligible for the purposes of our study.

Minor comments:

5. L61, Reference 9, “Wang, E. & Jeffery 2017”, looks out of place and not related to polyphenol production.

Answer:

Thank you for recognizing! This reference was falsely annotated and is now removed in this revised version of the manuscript.

6. L171, Fig. 3c y-axes labels 18:2n9 should be 18:2n6.

Answer:

Thank you for noticing! The labels of the y-axis in figure 3c have been updated.

7. L165, Fig. 5 caption scale bar 5 μ M likely should be μ m.

Answer:

Thank you! We have corrected the scale bar from 5 μ M to 5 μ m in the caption of figure 5 (line 285).

8. L224, Fig. 4. In panel b LA tPAL addition shows a response with 2.5 min but in panel a there is no effect at 0, this was very confusing until I read the associated main text “At the beginning of the cultivation (in absence of LA and tPAL)”, to avoid similar confusion by others maybe some tweak of the figure would be in order, e.g. add line or arrow (to indication FA addition) and/or append caption text to would avoid confusion.

Answer:

We agree! **To avoid any confusion, the following sentence was added** to the caption of figure 4a: “Laurdan GP values at 0 h give the generalized polarization at the beginning of the cultivation in absence of LA or tPAL.” (line 239).

9. L278, “overall neutral polar lipids 33–36.”. Some reference mix up e.g. reference 35 is the GROMACS MD package reference.

Answer:

Thanks for pointing this out” References 35 and 36 were indeed out of place and removed.

10. L278, “MD simulations demonstrated that RES has a high affinity to the surface of lipid bilayers of the *C. glutamicum* model membrane, which agrees well with previous results from model neutral lipid bilayers (Fig. 4c-d, Supplementary Fig.4)23,24,37–39.”. Fig. 4c or d do not show RAS membrane affinity and Fig. S4 kind of shows that, i.e. lot of RES in the membrane, but as the water phase is cut off that is not highlighted. RES has a very high water to lipid partitioning so this is not surprising but for the “MD simulations demonstrated that RES has a high affinity to the surface of lipid bilayers” statement maybe something like a density plot would be clearer?

Answer:

The mentioned references reveal the partitioning of RES at a membrane interface from various methods, experimental and theoretical, including modeling with the MARTINI force field that was used in our study. As the references are added at the end of the mentioned statement, they were intended to support the whole statement. However, we also provide a proof of this well known aspect in our study in Figure S4 (indeed, not in Figure 4), where it is clearly shown that all RES molecules reside at the interface of the membrane. No molecules except water molecules and ions were omitted in those figures.

To clarify this topic, we have changed the wording of the mentioned statement in the main text to “The available experimental and theoretical evidence demonstrated that RES has a high affinity to the surface of lipid bilayers in general (23,24,37–39), which agrees with our results for the *C. glutamicum* model membrane (Supplementary Fig.4).” [– now Supplementary Figure 5!] (line 297)

In the caption to Figure S4 (now S5), **we added a statement** that “*Snapshots show all molecules except water molecules and ions, which are omitted for clarity.*” before the description of the color code.

11. L597, “Identification and quantification of membrane lipids”. Was there any quantification of membrane lipids reported?

Answer:

Thank you for this comment. In line 643, we specified the GC method for identifying membrane lipids, which were described in the results section ‘Supplementation of fatty acids counteracts RES effects’.

12. L640, "Error bars were computed from two independent membrane leaflets." Does that mean range was used?

Answer:

The statement specifies in a short form that the two leaflets in the membrane were used as two independent measurements and error estimates were derived from them using standard formulae. We rephrased the sentence in the "Methods"-section: "*Error bars were computed considering both leaflets as independent measurements.*" (line 677).

13. The videos that come with the reviewer material were not labeled 1 to 4 name e.g. 405795_0_video_7205006_rncd25 and the quality was rather low, but the download links in the SI did work. For video 4 – without LA for clarity, I think that video also either is not showing or dose not have any RES?

Answer:

We are sorry for the quality of the videos in the reviewer material. We only can comment that the videos we have provided have sufficient resolution.

Thank you for catching the mistake in the caption of the video 4, it shall indeed specify that "LA and RES are omitted for clarity". The intention is to make comparison with video 1 (control) easier. **We have changed the caption to make this clear:** "*For more clarity and to ease the comparison with Movie 1, both RES and LA are not shown.*"

Reviewer #2:

This paper by Tharmasothirajan et al. proved the toxicity of polyphenol synthesis on the cell membrane and explored how adding free fatty acids can attenuate such toxicity and improve the production of polyphenols. In brief, using resveratrol (RES) as a representative polyphenol, the authors first analyzed its distribution and found out most RES is located in cell pellet with more fraction in cell membrane compared to cytoplasm. The authors then hypothesized that such a high concentration of RES would cause toxicity and inspired by previous studies, they attempted to add different free fatty acids to reduce the toxicity. Among the 12 tested fatty acids, most of them could optimize cell growth and the supplementation of palmitelaidic acid and linoleic acid most effectively increased the RES yield. Subsequently, the mechanism was elucidated by analyzing and modeling the structure and properties of the cell membrane. RES rigidifies the cell membrane and decreases its fluidity, and the incorporation of palmitelaidic acid and linoleic acid into the cell membrane increases its fluidity to counteract the effects of RES. Overall, this is a well-conducted study demonstrating the capability of free fatty acid supplements to relieve the toxicity of

polyphenols for higher production. However, I do not think it is novel or influential enough to be published on Nature Communications.

1. As the authors mentioned, the influence of free fatty acids on the cell membrane and tolerance resistance has been explored in many previous studies [1].

[1. Sandoval N R, Papoutsakis E T. Engineering membrane and cell-wall programs for tolerance to toxic chemicals: beyond solo genes. Current opinion in microbiology, 2016, 33: 56-66.]

Answer:

Here we disagree. We only wrote in the introduction that recent studies “*demonstrated that bacteria such as Enterococcus faecalis or pathogenic Vibrio species are able to incorporate exogenous fatty acids into membrane phospholipids, thereby altering their membrane composition to increase antibiotic resistance.*” These findings were made in a medical context. We clearly state that “*potentially beneficial effects of exogenously supplied FAs on microbial membrane characteristics in the context of biotechnological production processes have not been extensively studied.*” This is indeed the case. The excellent review by Sandoval and Papoutsakis the author focus on laboratory evolution/metabolic engineering approaches to alter microbial membranes. The only time free fatty acids are mentioned in this review is in the context of toxicity when the authors describe that “*Exogenous free fatty acids, common in biodiesel-derived crude glycerol feedstocks, can partition into membranes and have a detrimental effect on growth and fermentation.*”

2. It will be more convincing if the authors can use this approach to optimize previous overproducers and achieve higher titers than previous studies.

Answer:

Thank you for making this point, because this is what we did. *C. glutamicum* has been previously engineered for plant polyphenol / polyketide synthesis by us and others. The strains for RES and NAR production we use are production strains. Unfortunately, we do not have access to other organisms producing such compounds, but we already suggested our strategy to some of our colleagues, who see very similar effects on production and cell viability upon supplementation of free FAs.

The following comments hopefully would help the authors to further improve this paper.

3. Line 118, the authors might want to provide the data of cell growth under different concentrations of RES to fully prove the cytotoxicity of RES.

Answer:

In our previous work, the effect of different RES concentrations (0 - 4.4 mM) on growth of *C. glutamicum*-RES was thoroughly investigated to estimate cytotoxic effects of this

valuable polyphenol (Reference: Tharmasothirajan, A., Wellfonder, M. & Marienhagen, J. Microbial polyphenol production in a biphasic process. ACS Sustain. Chem. Eng. **9**, 17266–17275 (2021)). In this context, we could demonstrate that growth of *C. glutamicum*-RES was negatively affected in presence of RES concentrations ≥ 0.66 mM. Based on these results a IC_{50} of 2.6 mM for RES was calculated for *C. glutamicum*-RES.

This information is already included in the second paragraph of the discussion (line 412): *“In recent years, C. glutamicum emerged as promising host for the production of plant polyphenols such as stilbenoids and flavonoids and the impact of the stilbenoid RES on C. glutamicum-RES1 was investigated during growth studies in defined CGXII medium.”*

However, we upon the reviewer’s suggestion we decided to include this additional sentence for more clarity: *“Previous studies also demonstrated a dose-dependent inhibitory effect different RES concentrations on the growth of C. glutamicum. In the context of these experiments an IC_{50} of 2.6 mM could be determined for RES.”* (line 121)

4. Line 127-131, the related reference should be provided to support that supplementation of FAs can positively influence the membrane composition of some bacteria for enhanced antibiotic resistance.

Answer:

Thank you. The references were included in the introduction section, but it makes absolutely sense to also include them in the result section (line 136).

5. Appropriate references were absent, for example, line 128 and line 246.

Answer:

Thank you for noticing. Appropriate references were added in the revised manuscript (now line 136 and 265).

25. Moravec, A. R. et al. Exogenous polyunsaturated fatty acids impact membrane remodeling and affect virulence phenotypes among pathogenic *Vibrio* species. *Appl. Environ. Microbiol.* **83**, 1415–1417 (2017).

26. Saito, H. E., Harp, J. R. & Fozo, E. M. Incorporation of exogenous fatty acids protects *Enterococcus faecalis* from membrane-damaging agents. *Appl. Environ. Microbiol.* **80**, 6527–6538 (2014).

30. Golfetto, O., Hinde, E. & Gratton, E. Laurdan fluorescence lifetime discriminates cholesterol content from changes in fluidity in living cell membranes. *Biophys. J.* **104**, 1238–1247 (2013).

31. Parasassi, T. & Gratton, E. Membrane lipid domains and dynamics as detected by Laurdan fluorescence. *J. Fluoresc.* **5**, 59–69 (1995).

32. Malacrida, L., Jameson, D. M. & Gratton, E. A multidimensional phasor approach reveals LAURDAN photophysics in NIH-3T3 cell membranes. *Sci. Rep.* **7**, 1–11 (2017).

33. Vequi-Suplicy, C. C., Lamy, M. T. & Marquezin, C. A. The new fluorescent membrane probe Ahba: A comparative study with the largely used Laurdan. *J. Fluoresc.* **23**, 479–486 (2013).

34. Bonaventura, G. et al. Laurdan monitors different lipids content in eukaryotic membrane during embryonic neural development. *70*, 785–794 (2014).

35. Harris, F. M., Best, K. B. & Bell, J. D. Use of laurdan fluorescence intensity and polarization to distinguish between changes in membrane fluidity and phospholipid order. *Biochim. Biophys. Acta - Biomembr.* **1565**, 123–128 (2002).

6. Line 153-155, “In all cases, the growth behavior was similar to the EtOH control confirming

that *C. glutamicum*-RES cannot metabolize any of the supplemented compounds”, why did the addition of unsaturated trans-FAs increase the biomass formation by 14 %? Please provide the perspectives.

Answer:

Thank you very much. **Supplementary figure 1 illustrates** the growth behavior of *C. glutamicum*-Res on the supplemented FAs over time, showing **that the presence of trans-unsaturated FAs did not result in improved biomass formation**. This indicates that *C. glutamicum*-RES1 cannot metabolize FAs. However, it would be surprising anyhow, since *C. glutamicum* ATCC 13032 used in this study lacks the enzymatic machinery for β -oxidation. Furthermore, also in Figure 6, Supplementary Figure 2 and Supplementary Figure 3, no improved biomass formation of *C. glutamicum*-RES1 under similar cultivation conditions was observable (compared to EtOH-control). However, we agree that the representation in Figure 3a might hint towards a slightly increased final optical density of the cultures after 72 hours (and we mentioned this in the text ourselves (line 153) and subsequently performed the mentioned additional experiments to show that this is not the case).

*7. Based on these results, 40 μ M LA or 40 μ M tPAL appear to be sufficient for increased RES production using *C. glutamicum*-RES. To increase the resveratrol titer, LA or tPAL was added repetitively. Whether the resveratrol titer will further increase when FA and tPAL were added simultaneously and repetitively?*

Answer:

Interesting idea, thank you! To investigate this, **we performed additional shake flask cultivations in defined CGXII medium to examine the RES production in presence of LA – tPAL mixtures**. In Supplementary Figure 2, we could determine the optimal FA concentrations for increased RES production to be 40 μ M. Therefore, we selected a LA – tPAL mixture of 20 μ M LA + 20 μ M tPAL for a total FA concentration of 40 μ M. Out of curiosity, we also tested a 40 μ M LA + 40 μ M tPAL mixture to keep both FAs at their respective individual optimal concentration.

Compared to supplementation of 40 μ M LA or tPAL alone, addition of the 20 μ M LA+ 20 μ M tPAL resulted in the same final product titer of 1.8 mM RES. However, supplementation of the 40 μ M LA + 40 μ M tPAL mixture did not further improve the RES titer. **We expanded Supplementary Figure 3 to accommodate these results and added the information in the revised manuscript** in the results sections:

‘Supplementation of tPAL or LA improves microbial RES synthesis’

*“In order to explore the possibility of further increasing RES production in *C. glutamicum*-RES1, LA and tPAL were added simultaneously to the culture medium (Supplementary Fig. 3). An*

FA-mixture containing 20 μM LA and 20 μM tPAL (total FA concentration of 40 μM) and a mixture containing 40 μM LA and 40 μM tPAL (total FA concentration of 80 μM) were used. The addition of a 40 μM FA mixture resulted in the same final titer of 1.8 mM RES compared to the single supplementation with 40 μM LA or tPAL. However, the supplementation of the 80 μM FA mixture did not further improve the production performance.” (line 213).

Additionally, we tested the repetitive addition of both mixtures with no positive effect on the production performance compared to the single and repetitive supplementation of 40 μM LA or tPAL alone. Since no improvement was observed, we did not include this experiment in the revised manuscript.

8. Although the free fatty acids supplied in this study had been proven not involved in cell metabolism, is there any possibility that the added fatty acid were able to extract the products, thereby alleviating the cytotoxicity of the products?

Answer:

Thank you, this is a very interesting question, which we also discussed prior to the initial submission. We believe, that an *in situ* product removal by the supplemented FAs can be excluded because to the low FA concentrations used throughout our experiments. **FAs exhibit a critical micelle concentration (CMC) in the range of 60-100 μM** and the concentration of 40 μM typically used in our experiments is way below this CMC (*Reference: Serth, A., Lautwein, A., French, M., Wittinghofer, A., Pingoud, A. The inhibition of the GTPase activating protein Ha-ras interaction by acidic lipids is due to physical association of the C-terminal domain of the GTPase activating protein with micellar structures. EMCO J. 10, 1325-1330 (1991)*). Moreover, our **thin-layer chromatography (TLC) experiments showed that LA and tPAL are indeed present as free FAs in the membrane of *C. glutamicum*** (Supplementary figure 6), thus further decreasing the FA concentration in the culture medium. Additionally, the concentration of LA and tPAL used in our cultivation experiments (40 μM) can also regarded as being too low to extract 1.5 mM of RES from the culture broth. Taken together, under our experimental conditions, the extraction of RES through FA micelles can be excluded.

Nonetheless, we have added this important aspect to the discussion in the revised manuscript: “*In situ* extraction of RES by the FAs in the culture supernatant can be excluded due to the FAs concentration used being below the critical micelle concentration of 60-100 μM ” (line 466).

9. The authors talked a lot about how resveratrol and fatty acid alter cell membrane in detail. However, more explanations about how the changed cell membrane properties facilitate polyphenols need to be provided. For example, fatty acid increased the membrane fluidity, why the increased fluidity contributed to resveratrol production?

Answer:

In this study, we could show that **the membrane is influenced by RES, resulting in increased membrane rigidity, since RES interacts with membrane components** (Figure 4a+4b, Supplementary Figure 4). Therefore, RES can only pass through the membrane to a limited extent. **Laurdan kinetic measurements demonstrated improved membrane fluidity during RES production in presence of LA or tPAL compared to the control culture without FA supplementation** (Figure 4b). Due to increased fluidity, RES can pass through the membrane more easily. In turn, we assume that a reduced intracellular concentration of the eventually toxic RES has a positive effect on the metabolism of *C. glutamicum*, which is apparently reflected in improved RES synthesis. **The finding of our study is that FA supplements help in restoring the membrane physical state that is altered by RES production and RES accumulation.**

10. If the increased membrane fluidity could increase export across the membrane, more products will go to the supernatant. However, more increased production was not accumulated in the supernatant according to Fig.6. When producing naringenin, it even looks like no increase in the supernatant.

Answer:

Thank you very much for noticing. We went back to the original raw data obtained from these cultivations with multiple FA supplementations as this is indeed not in line with the numerous experiments conducted with single supplementations. The data visualization in the revised version has been corrected based on the recalculation and the source data file has been updated. We performed the experiments again, and received the same results. We can only apologize.

Noteworthy, RES, like all stilbenoids, is known for its poor water solubility. **Our experiments show that only 1.5 mM of RES can be dissolved in defined CGXII medium** at 30°C and pH 7, which explains the low concentration observed in the culture supernatant (see below).

We can only assume that excess RES precipitates in the culture medium. In a previous study, we established a biphasic cultivation process for *C. glutamicum*-RES1 producing RES, in which RES is extracted into the organic phase during the cultivation process to overcome RES precipitation.

[1] Tharmasothirajan, A., Wellfonder, M. & Marienhagen, J. Microbial polyphenol production in a biphasic process. ACS Sustain. Chem. Eng. **9**, 17266–17275 (2021).

11. To better connect cell membrane properties with resveratrol production, more control groups are needed. For example, adding certain reagents rather than fatty acids to change the cell membrane fluidity and measuring the effect on resveratrol production.

Answer:

In initial experiments, we also evaluated permeabilizing detergents, such as Tween40, Tween80, Ethambutol, for a positive effect on RES production with *C. glutamicum*-RES1. However, the above-mentioned detergents had a pronounced negative effect on the viability of *C. glutamicum*-RES1 and no improvement in RES production could be observed.

Investigation of permeabilizing detergents on growth and RES-production of *C. glutamicum*-RES1. a Biomass formation and b RES production in *C. glutamicum*-RES1 in presence of 1% (v/v)

Tween40, Tween80 and 500 mg/L ethambutol. Data represent average values and standard deviations of three biological replicates.

Due to the restrictions of the text-length and in order to keep the focus, we did not include these findings in the final manuscript.

However, we already did mention these detergents in the discussion, when discussing attempts to weaken the cell envelope in the context of L-glutamate export (line 445).

12. Fig.3, for the EtOH control group, not only the biomass formation and the resveratrol titer increased, but also the product titer ratio of total RES vs extracellular RES changed, just similar to many FA groups, may EtOH also have effects on membrane characteristics? If EtOH can act on membrane characteristics, how can EtOH impacts be excluded for the experimental groups?

Besides, what's the relationship between Fig 3b and 3c, from my understanding, final product titers were normalized to optical density (OD₆₀₀) to get Fig. 3c, but why only the control group in Fig 3c shows evident differences in the ratio of total RES vs extracellular RES compared to that of 3b?

Answer:

C. glutamicum has the metabolic capability to utilize EtOH as carbon and energy source. Therefore, the control cultivation supplemented with EtOH only showed a significant increase in biomass formation compared to the control group without EtOH (Figure 3a). **Furthermore, EtOH is oxidized via acetate to acetyl-CoA as direct precursor of malonyl-CoA, which is an important precursor for RES production and thus contributes to the increase in RES production in EtOH control experiments** (see Figure 1). The Laurdan assays conducted in this study demonstrated that small amounts of 0.05% EtOH do not have any impact on membrane fluidity of *C. glutamicum* with a OD₆₀₀ of 0.5 (0.17 g/L biomass).

Since all tested FAs are stored in EtOH, a control group with the same final concentration of EtOH as the culture supplemented with FAs was included in all conducted experiments to always take the influence of EtOH on RES production into account. All cultures supplemented with FAs were characterized by biomass formation similar to the EtOH control, but a significantly increased RES production solely due to the presence of FAs.

The RES production in *C. glutamicum* is dependent on the growth of the microorganism as it is linked to biomass formation via RES-precursor synthesis. To facilitate a more accurate comparison to the impact of each individual FA supplement, the final product titers were always normalized to the final optical density (OD₆₀₀), resulting in individual product yields (mM/OD₆₀₀) shown in Figure 3c.

13. Page 8, "In all cases the growth behavior was similar to the EtOH control confirming that *C. glutamicum*-RES cannot metabolize any of the supplemented compounds." And in page 20, the paper goes that "Our data show that LA and tPAL are only incorporated into phospholipids of *C. glutamicum* at low quantities, but remain in the cell envelope as free FAs." So the supernatant FAs and total FAs titers are suggested to be detected and added, this will help support these two conclusions.

Answer:

Supplementary Fig. 6 presents LC-MS data qualitatively confirming the presence of LA and tPAL as free FAs in the membrane of *C. glutamicum*, when cultivated in presence of these FAs. Given the very low concentrations of the FAs used, we do not think that the free FAs can be reliably quantified. **Furthermore, the residual free FAs in the supernatant and the FAs in the membrane (free or incorporated into phospholipids) would never add up to 40 μ M.**

14. There are many grammar issues in this paper. As an example, in line 63, do you mean elucidation by "elimination"? I also do not think "here" is appropriate. Lines 83 and 201 also have grammar issues. The authors should revise the paper carefully.

Answer:

Thank you for your comment. Our native speakers among the authors have reviewed the text again and made corrections. In total, 29 corrections were made in the manuscript and one correction in the supplementary information. All changes are highlighted in yellow.

Reviewer #3:

*This manuscript describes production of resveratrol (RES) by engineered *Corynebacterium glutamicum*. The authors showed that supplementation of palmitelaidic acid (tPAL) and linoleic acid (LA) improved production and extracellular export of RES. By supplementation of 40 μ M LA, 2.47 mM RES was produced from 5 mM p-cumaric acid, which was 2.8-fold higher than that of the control. The authors also showed that supplementation of tPAL and LA increased and restored the membrane fluidity rigidified by RES synthesis, although the membrane composition was not altered by supplementation of tPAL and LA.*

1. In the aspect of basic research, it is interesting that supplementation of the fatty acids improved production and export of RES, but no mechanism was described in this manuscript.

Answer:

Here we strongly disagree and can only refer the reviewer to the text. Our experiments show that supplemented FAs are incorporated into the membrane as free FAs and alter the fluidity and lateral organization within the membrane, which is more rigid due to the accumulation of RES. The observed changes in fluidity and lateral organization upon FA supplementation results in the restoration of the membrane to its original state, eventually increasing viability and product efflux. This was demonstrated by cultivations/chemical analyses, Laurdan assays, fluorescence microscopy and state-of-the-art molecular dynamics simulations.

2. Besides, it is already known that the supplementation of fatty acids influences the cellular physiology by changing the membrane composition as stated in Introduction (P5 L86).

Answer:

Here we disagree. We only wrote in the introduction that recent studies “*demonstrated that bacteria such as Enterococcus faecalis or pathogenic Vibrio species are able to incorporate exogenous fatty acids into membrane phospholipids, thereby altering their membrane composition to increase antibiotic resistance.*” These findings were made in a medical context. (line 87). These studies also show that the supplemented FAs are incorporated into membrane phospholipids. In contrast, we clearly state that “*potentially beneficial effects of exogenously supplied FAs on microbial membrane characteristics in the context of biotechnological production processes have not been extensively studied.*” (line 90). In our study, we could show for the first time that free FAs are incorporated into the membrane as free FAs alter the fluidity and lateral organization within the membrane, which is more rigid due to the accumulation of an aromatic and hydrophobic product.

3. In the aspect of applied research, it is important to produce industrially important chemicals from inexpensive sugar-based feedstocks is important as stated in Introduction (P3 L58). However, in this manuscript, RES was produced from p-coumaric acid.

Answer:

Thank you for this suggestion. To investigate the influence of FAs on RES production and product export across the membrane of *C. glutamicum*, the supplementation of *p*-coumaric acid was deliberately chosen for a better comparison between cultivations, as cultivations with this precursor molecule allow for higher titers. This is a common approach during metabolic engineering campaigns to be able to observe the impact of individual engineering steps on the overall product titers. Furthermore, RES is a rather expensive and pharmaceutically relevant compound instead of a bulk-product. Hence, the price of the substrate is not really relevant.

However, to also demonstrate that LA and tPAL supplementation has a positive effect on microbial RES production starting from sugar-based feedstocks, **we performed additional experiments in which RES was produced from D-glucose as sole carbon- and energy source** using *C. glutamicum*-RES2. This strain also harbors the gene for the 3-deoxy-D-arabino-heptulosonate-7-phosphate (DAHP) synthase (AroH) from *E. coli* and a codon optimized version of the gene for tyrosine ammonia lyase (TAL) originating from *Flavobacterium johnsoniae*. Both genes are plasmid-encoded, enabling the synthesis of *p*-coumaric acid from L-tyrosine.

Conducted cultivation experiments with this strain confirm that supplementation of LA and tPAL allow for a 3-fold increased RES production – also from D-glucose.

We decided to include these results into the manuscript, and expanded the scheme of the metabolic pathway for RES synthesis from D-glucose in Figure 1 and added details regarding the strain *C. glutamicum*-RES2 in the “Materials”-section of the revised manuscript. Results of the performed cultivations of this strain in the presence of LA or tPAL were also included in the “Results”-section:

“Finally, it was evaluated whether supplementation of LA or tPAL has also a positive impact on microbial RES synthesis starting from cheap D-glucose instead of the RES-precursor p-CA. Therefore, C. glutamicum-RES2 was constructed, which was engineered for the additional heterologous expression of genes for the 3-deoxy-D-arabino-heptulosonate-7-phosphate (DAHP) synthase (AroH) from E. coli and the codon optimized tyrosine ammonia lyase (TAL) originating from Flavobacterium johnsoniae in comparison to C. glutamicum-RES1. These two additional enzymatic activities enable the overproduction of L-tyrosine and the conversion of L-tyrosine to p-CA. Cultivation of this strain variant in the presence of 40 µM LA or tPAL allowed for 3-fold increased RES accumulation and export across the membrane of C. glutamicum-RES2 in comparison to suitable controls. These results underline the positive effect of FA supplementation on microbial plant polyphenol synthesis (Supplementary Figure 7).”

An additional figure (Supplementary Figure 8) was added to the Supplementary Information.

4. Besides, the production titer and yield was not prominent compared to the previous studies on RES production (PMID: 30798357).

Answer:

Thank you for this comment. The excellent review by Shrestha et al. gives a good overview of the past engineering efforts towards microbial production of different stilbenoids. **Table 1 in this review gives an overview of titers reached with different organisms. However, what is completely lacking is any comparison of:**

- **cultivation conditions/process parameters (starting biomass?, growing/resting cells?, substrate concentration(s)?, cultivation time?)**
- **concentration of the supplemented precursor(s) (if any)?**
- **supplementation of very expensive growth inhibitors such as cerulenin to boost malonyl-CoA availability?**

Hence, the **information regarding the strain performances in the review mentioned by Reviewer #3 cannot be compared at all**. We are a lab working on *Corynebacterium glutamicum* for various applied purposes. The **novelty of this study** is not “the highest titer or yield” – but **the finding that supplementation of small amounts of cheap FAs increases cell viability and boosts overall product formation**. We hope that our findings will motivate colleagues working on the same products but with different host system to also try this new strategy. For additional information, we would direct you to comment 6 of Reviewer #3, which is basically the same point.

Major comments:

5. P16 L323 *How do free LA and tPAL increase the membrane fluidity without changing the membrane composition? We need to know the mechanism.*

Answer:

This is the same comment as comment #1 of the same reviewer: Here we strongly disagree and can only refer the reviewer to the text. Our experiments show that supplemented FAs are incorporated into the membrane as free FAs alter the fluidity and lateral organization within the membrane, which is more rigid due to the accumulation of RES. The observed changes in fluidity and lateral organization upon FA supplementation results in the restoration of the membrane to its original state, eventually increasing viability and product efflux. This was demonstrated by cultivations/chemical analyses, Laurdan assays, fluorescence microscopy and state-of-the-art molecular dynamics simulations.

6. P17 L344 *It is high titer of RES by LA supplementation (2.47 mM (564 mg/L)) for C. glutamicum as a host strain, but there is a paper reporting production of higher titer of RES, 1.4 g/L from p-cumaric acid and 2.3 g/L from p-cumaric acid and cerulenin with high yield using E. coli (PMID: 21441338). What is the advantage of using C. glutamicum for RES production?*

Answer:

We would also like to refer any reader to comment 4 of Reviewer #3 and the our answer, since this comment is essentially the same.

Our study cannot be compared to the mentioned excellent study of Mattheos Koffas at all. This *E. coli* strain was cultivated under completely different conditions, which are unsuitable for any large scale production condition. In order to achieve a final product titer of 1.4 g/L, our colleagues used:

- an initial OD₆₀₀ of 20-25
- starting concentration of 15 mM *p*-coumaric acid

Product titers of 2.3 g/L were only reached with supplementation of **cerulenin** at a final concentration of 50 μM (**50 mg of this fatty acid inhibitor need to be purchased for 1,490 EUR (1,630 USD)**, MERCK, price as of April 25th). This alone would prohibit any future commercialization.

In contrast, we used:

- a **five-fold lower inoculum** (OD₆₀₀ = 5)
- a **three-fold lower initial *p*-coumaric acid concentration of 5 mM**

to be able to work **with growing cells**.

7. P20 L425 Is there a correlation between the effect of RES on decreasing the membrane fluidity and cytotoxicity? Please explain.

Answer:

In our previous work, the effect of different RES concentrations (0 - 4.4 mM) on growth of *C. glutamicum*-RES was thoroughly investigated to estimate cytotoxic effects of this valuable polyphenol (*Reference*: Tharmasothirajan, A., Wellfonder, M. & Marienhagen, J. Microbial polyphenol production in a biphasic process. ACS Sustain. Chem. Eng. **9**, 17266–17275 (2021)). In this context, we could demonstrate that growth of *C. glutamicum*-RES was negatively affected in presence of RES concentrations ≥ 0.66 mM. Based on these results, a IC₅₀ of 2.6 mM for RES was calculated for *C. glutamicum*-RES.

This information is already included in the second paragraph of the discussion (line 412): *“In recent years, C. glutamicum emerged as promising host for the production of plant polyphenols such as stilbenoids and flavonoids and the impact of the stilbenoid RES on C. glutamicum-RES1 was investigated during growth studies in defined CGXII medium.”*

However, we upon the reviewer’s suggestion we decided to include this additional sentence for more clarity: *“Previous studies also demonstrated a dose-dependent inhibitory effect different RES concentrations on the growth of C. glutamicum. In the context of these experiments, an IC₅₀ of 2.6 mM could be determined for RES.”*

With the aim to demonstrate the immediate effect of RES on the cell membrane of *C. glutamicum*-RES in more detail, **we conducted additional Laurdan kinetic measurements in the presence of different RES concentrations ranging from 0.13 - 0.75 mM**. The correlation between RES-induced decrease in membrane fluidity and cytotoxicity is further

supported by the observed reduction in membrane fluidity with increasing RES concentrations in these experiments. **We included these results as new Supplementary Figure 4.**

8. P20 L433 How do LA and tPAL in cell envelope counteract the cytotoxic effect of RES in cytoplasm without incorporated in the cell membrane? Please explain.

Answer:

A possible explanation for this was found by our molecular dynamics simulations: MD simulations show that RES significantly increases the rigidity of lipid membranes while slowing down their dynamics. This is reflected in the membrane lateral compressibility coefficient as measure for the elasticity of lipid membranes, which increases two-fold upon adding RES to a RES:lipid ratio of 1:2 (Fig. 4c, Supplementary Fig.5). At the same time, the diffusion coefficient of POPS as the dominant lipid in *C. glutamicum* membranes reveals an approximately two-fold slowdown. The membrane rigidifying effect of RES is presumably linked to an alteration in the lateral organization of the membrane. The addition of RES brings the most abundant POPS lipids in closer proximity and increases their segregation with the other lipids, causing the formation of transient nanoscale lipid domains.

These experimentally observed positive effects of tPAL and LA supplementation on the membrane fluidity of *C. glutamicum* are also supported by MD simulations of *C. glutamicum* model membranes. Here, addition of LA or tPAL to the simulations in similar amounts to RES, increased lipid diffusion rates and membrane fluidity in both cases. **In some simulations, FA supplementation even overcompensated the negative RES effects resulting in model membranes with increased lipid diffusion and membrane fluidity (Fig. 4c+d, Supplementary Fig. 5, Supplementary Movie 3+4).** Furthermore, LA or tPAL also restored the lateral organization of the membrane lipids suppressing the formation of the previously observed RES-induced nanoscale lipid domains.

9. P20 L434 If the supplementation of LA and tPA just restore the original cell envelop properties, how does it improve resveratrol production and export? Please explain.

Answer:

In this study, we could show that the membrane is influenced by RES, resulting in increased membrane rigidity, since RES interacts with membrane components (Figure 4a+4b, Supplementary Figure 4). Therefore, RES can only pass through the membrane to a limited extent. **Laurdan kinetic measurements demonstrated improved membrane fluidity during RES production in presence of LA or tPAL** compared to the control culture without FA supplementation (Figure 4b). **Due to increased fluidity, RES can pass through the membrane more easily.** In turn, we assume that a reduced intracellular concentration of the

toxic RES has a positive effect on the metabolism of *C. glutamicum*, which is apparently reflected in improved RES synthesis. The finding of our study is that FA supplements help in restoring the membrane physical state that is altered by RES production and RES accumulation.

Briefly, the membrane environment is affected by RES towards more rigid state. FAs help restoring the membrane state back closer to the optimum.

Minor comments

10. P8 L160 In Fig. 3b, it looks like there is clear difference between extracellular and total RES concentrations on supplementation of palmitelaidic, linoleacdic, and linoleic acids.

Answer:

Thank you for noticing. The difference between extracellular RES and total RES concentration upon supplementation of *palmitelaidic (tPAL)*, *linoleacdic*, and *linoleic acid (LA)* can be explained by the poor water solubility and high affinity to hydrophobic solutions. Experimental data demonstrate that only up to 1.5 mM RES can be dissolved in defined CGXII medium at 30°C and pH 7, which explains the observed difference between extracellular and total RES concentrations. RES concentrations in all supernatant samples were nearly the same.

11. P8 L165 No data for LA supplementation in Fig. 3C.

Answer:

Thank you for this comment. LA was falsely annotated as 18:2n9 instead of 18:2n6. The labels on the y-axis in Figure 3c have been corrected. Hence, the data for LA is present in Figure 3c.

12. P10 L193 In supplementary Fig. 2, 40 µM is the best concentration for RES production? How about 20 or 30 µM?

Answer:

Very good point, thank you! Indeed, for the sake of completeness, the impact of 20 µM and 30 µM of LA as well as tPAL on RES production should have been investigated. Corresponding **stock solutions for final concentration of 20 µM and 30 µM LA and tPAL in defined CGXII medium were ordered and tested in additional experiments.** The results indicate that FA concentrations below 40 µM does not have a positive effect on RES production in *C. glutamicum*. **These data have been added to Supplementary Figure 2.**

Reviewers' Comments:

Reviewer #1:

Remarks to the Author:

I do think the revised manuscript overall is improved and think it is of significant interest to the scientific community. That being said I was not satisfied with some of the answers to my questions. I will focus on the most important one: Question 1 has been misunderstood or dismissed so I will go into it in more detail.

The effect of RES on membranes has been studied before, and RES identified as belonging to a group of several other phytochemicals which look to have similar membrane "softening" effects, at least in simple membranes at low concentrations. Even though the RES rigidifying effect is not the main point of this paper, it is a prominent point, and you will be publishing that RES is rigidifying to membranes in a prestigious and widely read journal, therefore, not addressing why you see different results, not mentioning that they are different, or much worse with the new addition "Interestingly, previous MD studies probing the effect of RES on membrane properties concluded a less perturbing effect, which we attribute to the lower concentrations of RES (~10 mol%) compared to our study (50 mol%)^{23,42,43.}" incorrectly quoting previous work, which do show an effect and it makes it easier to bend the membrane which clearly is not rigidifying.

The three studies I provided last time:

<https://pubs.acs.org/doi/10.1021/cb500086e>

<https://pubs.acs.org/doi/10.1021/acs.jcim.2c00372>

<https://dx.doi.org/10.1021/acs.jmedchem.0c00958>

indeed support that RES softening of the membrane via simulations and I am shocked by the authors' incorrect dismissal of them by picking out a few select analyses with high error bars. If you intend to draw contrary conclusions than the authors themselves in these papers based on their data, you should have a very good reason to do so and explicitly state them. Personally, I find the combined simulations evidence in those three papers stronger than the Martini simulation RDF and compressibility (without checks for changes in undulations) analyses evidence provided here supporting the rigidifying effects. But I also think both can be true depending on concentration and/or membrane environment.

Driving this home <https://pubs.acs.org/doi/10.1021/acs.jmedchem.0c00958> even quantifies the effect of RES via the gA-based perturbation assay, see Table 1, and <https://pubs.acs.org/doi/10.1021/cb500086e> shows clear effect of RES in an artificial bilayer perturbations assay e.g. see Fig. S2C. Even more importantly than the simulations results both <https://dx.doi.org/10.1021/acs.jmedchem.0c00958> and <https://pubs.acs.org/doi/10.1021/cb500086e> are backed up with solid experimental evidence – and actually the simulation portion of <https://pubs.acs.org/doi/10.1021/cb500086e> is the minor portion in support of the experimental evidence using a number of different systems and under different condition all showing RES behaving similar to other bilayer softening drugs.

Additionally, a number of membrane proteins are effected by RES, in a similar manner as other know membrane modulators (which all would be deemed to soften rather than rigidify membranes), see table 1 and table S1 in <https://pubs.acs.org/doi/10.1021/cb500086e> and references therein.

This publication also supports a decrease in membrane fluidity:

<https://pubmed.ncbi.nlm.nih.gov/20691168/>

Interestingly <https://pubmed.ncbi.nlm.nih.gov/23805753/> (and other publications from the same group) show poly-unsaturated fatty acids and have "similar" bilayer softening effects as measured using gA channel activity as RES <https://pubs.acs.org/doi/10.1021/cb500086e>. Given the reversal seen here this indicates either there is a biphasic effect based on RES concentration and/or PG lipids

change the effect of RES in some way.

I urge the authors to read the papers provided and correct their misquotation in manuscript. I do realize phytochemicals effect on bilayer properties is quite a quagmire and not urging anyone down that road but please don't add to the confusion.

Reviewer #2:

Remarks to the Author:

The manuscript has been revised well.

Reviewer #3:

Remarks to the Author:

The authors appropriately answered all points that the reviewer pointed out. In addition, new experiments were performed and appropriate figures were added. Therefore, the reviewer has no further comment.

Point-by-point response to Reviewers

Membrane manipulation by free fatty acids improves microbial plant polyphenol synthesis

Reviewer #1

1. I do think the revised manuscript overall is improved and think it is of significant interest to the scientific community. That being said I was not satisfied with some of the answers to my questions. I will focus on the most important one: Question 1 has been misunderstood or dismissed so I will go into it in more detail.

Reply: We are glad to read that the present work is considered of significant interest to the scientific community. To our knowledge, we provided answers to all questions, and reactions to all comments in the previous review round. In this review round, we will elaborate more on the topic selected by Reviewer 1.

2. The effect of RES on membranes has been studied before, and RES identified as belonging to a group of several other phytochemicals which look to have similar membrane “softening” effects, at least in simple membranes at low concentrations. Even though the RES rigidifying effect is not the main point of this paper, it is a prominent point, and you will be publishing that RES is rigidifying to membranes in a prestigious and widely read journal, therefore, not addressing why you see different results, not mentioning that they are different, or much worse with the new addition “Interestingly, previous MD studies probing the effect of RES on membrane properties concluded a less perturbing effect, which we attribute to the lower concentrations of RES (10 mol%) compared to our study (50 mol%)^{23,42,43.}” incorrectly quoting previous work, which do show an effect and it makes it easier to bend the membrane which clearly is not rigidifying.

The three studies I provided last time: <https://pubs.acs.org/doi/10.1021/cb500086e>, [1] <https://pubs.acs.org/doi/10.1021/acs.jcim.2c00372>, [2] <https://dx.doi.org/10.1021/acs.jmedchem.0c00958>, [3] indeed support that RES softening of the membrane via simulations and I am shocked by the authors' incorrect dismissal of them by picking out a few select analyses with high error bars. If you intend to draw contrary conclusions than the authors themselves in these papers based on their data, you should have a very good reason to do so and explicitly state them. Personally, I find the combined simulations evidence in those

three papers stronger than the Martini simulation RDF and compressibility (without checks for changes in undulations) analyses evidence provided here supporting the rigidifying effects. But I also think both can be true depending on concentration and/or membrane environment.

Reply: Concerning this point, we must disagree with Reviewer 1. Namely, **we do not find our conclusions in any conflict with previously published works**. In our perception, our statement in the manuscript that "*... previous MD studies probing the effect of RES on membrane properties concluded a less perturbing effect ...*" acknowledges that there is an effect of RES on the bilayer properties found in previous studies. It also acknowledges that there are differences between the observed effects in those studies and our work. The effects of RES on membrane physical properties do not depend only on the concentration of RES, but also on the target membrane composition and its physical state. The membrane compositions studied in the provided literature sources [3, 2, 1, 4] use only very simple single component phosphatidylcholine bilayers with various lipid tails. In comparison, **the membrane composition used in simulations in our work uses different lipids (no phosphatidylcholine) and it is substantially more complex**. In particular, we use a composition of 50% POPG, 25% DPPI, 12.5% PODG and 12.5% (PO)MGDG that was selected as a model membrane for *C. glutamicum* plasma membrane.

In our lab experiments, we have used fluorescence measurements of generalized polarization of a Laurdan probe obtained directly from the plasma membrane of *C. glutamicum*. Generalized polarization is a widely accepted fluorescence method that reports on the fluidity of the lipid matrix. For instance the publication [5] reports on the effects of cholesterol concentration on lipid bilayer properties using Laurdan generalized polarization (e.g. Figure 7 therein).

In our work, we report an increasing value of Laurdan generalized polarization from *C. glutamicum* plasma membranes after adding RES concentration, which denotes a rigidifying effect of RES on the membrane. Consistently, the generalized polarization drops with the addition of well-known fluidifying agents (Figure 4 in our manuscript). The findings from these experimental measurements are confirmed by our simulations that use a membrane composition modeling the lipid matrix of the plasma membrane of *C. glutamicum* and show qualitative agreement with the rigidifying effects observed in experiments.

In conjunction with this argument, we have adapted our manuscript by changing the following statement from:

"Interestingly, previous MD studies probing the effect of RES on membrane properties concluded a less perturbing effect, which we attribute to the lower concentrations of RES (10 mol%) compared to our study (50 mol%)^{23,42,43.}"

to:

*“Interestingly, previous MD studies probing the effect of RES on membrane properties concluded a less perturbing effect while experimental assays even point to a fluidizing effect, in line with the general action of related phytochemicals^{23,42,43}. We attribute this apparent discrepancy to the lower concentrations of RES (typically ~10 mol%) compared to our study (50 mol%), together with the difference in membrane composition. Most of the previous published results are based on model lipid membranes, whereas our results are obtained with a realistic multi-component mixture representing the plasma membrane of *C. glutamicum*.*

We think that in this way, the statement acknowledges the effects of RES in previous studies and clearly highlights their difference from our manuscript.

3. Driving this home <https://pubs.acs.org/doi/10.1021/acs.jmedchem.0c00958> even quantifies the effect of RES via the gA-based perturbation assay, see Table 1, and <https://pubs.acs.org/doi/10.1021/cb500086e> shows clear effect of RES in an artificial bilayer perturbations assay e.g. see Fig. S2C.

Even more importantly than the simulations results both <https://dx.doi.org/10.1021/acs.jmedchem.0c00958> and <https://pubs.acs.org/doi/10.1021/cb500086e> are backed up with solid experimental evidence – and actually the simulation portion of <https://pubs.acs.org/doi/10.1021/cb500086e> is the minor portion in support of the experimental evidence using a number of different systems and under different condition all showing RES behaving similar to other bilayer softening drugs.

Additionally, a number of membrane proteins are effected by RES, in a similar manner as other know membrane modulators (which all would be deemed to soften rather than rigidify membranes), see table 1 and table S1 in <https://pubs.acs.org/doi/10.1021/cb500086e> and references therein.

Reply: We do not doubt the validity of those results, and have indicated in our revised text that experimental assays point to a fluidizing role of RES and other phytochemicals (see response above). Note, however, that in [1], the reported MD simulations in fact show NO significant effect on the area compressibility upon addition of 10% RES to a POPC bilayer (Figure S1C).

4. This publication also supports a decrease in membrane fluidity: <https://pubmed.ncbi.nlm.nih.gov/20691168/> [4]

Reply: The paper mentioned here deals with the effect of RES on gel phase membranes, which is irrelevant for our study.

5. Interestingly <https://pubmed.ncbi.nlm.nih.gov/23805753/> (and other publications from the same group) show polyunsaturated fatty acids and have “similar” bilayer softening effects as measured using gA channel activity as RES <https://pubs.acs.org/doi/10.1021/cb500086e>. Given the reversal seen here this indicates either there is a biphasic effect based on RES concentration and/or PG lipids change the effect of RES in some way.

I urge the authors to read the papers provided and correct their misquotation in manuscript. I do realize phytochemicals effect on bilayer properties is quite a quagmire and not urging anyone down that road but please don't add to the confusion.

Reply: As already stated above, **our main results on the rigidifying effect of RES in bacterial membranes are from experimental measurements.** Our simulations with a multi-component model of the bacterial plasma membrane support this finding and provide molecular-level details about the effects of RES. Consistently both experiments and simulations also show the fluidifying effect of the fatty acid supplements used in our study. **In summary, both rigidifying as well as fluidifying effects are consistently revealed by our methods.**

To further clarify the origin of the differences seen in previous assays and our current one, we performed additional simulations of RES in simple POPC membranes. We find that RES has no clear effect on the area compressibility at 10 mol% (in line with the results reported in [1]), and also no effect at 50 mol%. These results suggest it is the composition of the membrane which causes the rigidifying effect of RES in the plasma membrane of *C. glutamicum*. These new results are included in the revised paper in the Supplementary Information as Supplementary Table 1. We added the following sentence to the main text:

*“To shed further light on this issue, we performed additional simulations of RES in a POPC model membrane, at 0, 10 and 50 mol%. We obtain area compressibilities that are rather insensitive to RES concentration (Supplementary Table 1), suggesting it is the particular lipid composition that is responsible for the observed rigidifying effect of RES in *C. glutamicum*.”*

We hope that this contributes to solving some of the apparent controversies.

Reviewers' Comments:

Reviewer #1:

Remarks to the Author:

I find that the last controversy has been resolved in the revised version, both with the more in-depth text and very well-illustrated with the added control simulations.

I do think the statement "Interestingly, previous MD studies probing the effect of RES on membrane properties concluded a less perturbing effect", as pointed out by the authors, the "less" is true for bilayer area compressibility but likely not true for some other measured properties such as change in lateral pressure profile (Fig 1) and bilayer local deformation (Fig S2c) in

<https://pubs.acs.org/doi/10.1021/cb500086e>. This likely reflect the complexity of the different energetic terms included in "membrane properties", the observed effect is clearly different and with the added discussion in the same paragraph I don't think will be misunderstood by future readers.

Point-by-point response to Reviewers

Membrane manipulation by free fatty acids improves microbial plant polyphenol synthesis

Editorial Requests

We addressed all editorial requests listed in the author checklist. As requested, we answered all points in the checklist and uploaded this list along with a revised version of the manuscript and all other necessary files and documents.

Reviewer #1

I find that the last controversy has been resolved in the revised version, both with the more in-depth text and very well-illustrated with the added control simulations.

I do think the statement “Interestingly, previous MD studies probing the effect of RES on membrane properties concluded a less perturbing effect”, as pointed out by the authors, the “less” is true for bilayer area compressibility but likely not true for some other measured properties such as change in lateral pressure profile (Fig 1) and bilayer local deformation (Fig S2c) in <https://pubs.acs.org/doi/10.1021/cb500086e>. This likely reflect the complexity of the different energetic terms included in “membrane properties”, the observed effect is clearly different and with the added discussion in the same paragraph I don’t think will be misunderstood by future readers.

Reply: Thank you! We are grateful for the points raised by all reviewers and constructive discussion. With the comments, questions and suggestions the manuscript could be greatly improved.